# Antidistillation Fingerprinting

**Yixuan Even Xu** [1]   **John Kirchenbauer** [2]   **Yash Savani** [1]   **Asher Trockman** [†1]
**Alexander Robey** [1]   **Tom Goldstein** [2]   **Fei Fang** [1]   **J. Zico Kolter** [1]

## Abstract

Model distillation enables efficient emulation of frontier large language models (LLMs), creating a need for robust mechanisms to detect when a third-party student model has trained on a teacher model's outputs. However, existing fingerprinting techniques that could be used to detect such distillation rely on heuristic perturbations that impose a steep trade-off between generation quality and fingerprinting strength, often requiring significant degradation of utility to ensure the fingerprint is effectively internalized by the student. We introduce *antidistillation fingerprinting* (ADFP), a principled approach that aligns the fingerprinting objective with the student's learning dynamics. Building upon the gradient-based framework of *antidistillation sampling*, ADFP utilizes a proxy model to identify and sample tokens that directly maximize the expected detectability of the fingerprint in the student after fine-tuning, rather than relying on the incidental absorption of the untargeted biases of a more naive watermark. Experiments on GSM8K, OASST1, and MBPP demonstrate that ADFP achieves a significant Pareto improvement over state-of-the-art baselines, yielding stronger detection confidence with minimal impact on utility across mathematical reasoning, dialogue, and code generation, even when the student model's architecture is unknown.

## 1. Introduction

Frontier large language model (LLM) development requires the investment of substantial computational resources available to very few institutions. As a result, modern foundation models are enticing targets for *model distillation*, where a *student* model is fine-tuned on the *teacher* model's outputs to replicate its capabilities at a fraction of the original training cost. Consequently, model owners may desire the ability to deploy *model fingerprinting* technologies that enable them to definitively determine whether a third-party model has been fine-tuned on the outputs of their model.

Recent research suggests that text watermarking techniques, such as the red-and-green-list scheme (Kirchenbauer et al., 2023), could be repurposed for this fingerprinting task (Sander et al., 2025). These methods rely on *heuristic* perturbations: they statically bias the sampling distribution toward a set of randomized "green" tokens. If such bias is present in the teacher, it may be learned by the student, enabling distillation to be detected. However, this simple approach makes perturbations to teacher outputs without considering how the student updates its parameters on these tokens. Empirically, these approaches impose a steep trade-off between generation quality and fingerprinting strength, often requiring significant degradation of the teacher to ensure the fingerprint is effectively internalized by the student.

In this work, we develop a more principled fingerprinting technique on top of the *antidistillation sampling* framework (Savani et al., 2025). The original antidistillation sampling approach implants specific functional biases into a student model via gradient-based logit perturbations derived from an auxiliary *proxy* model. While initially proposed as a defense that would degrade distilled student model's performance, we repurpose the core mechanism to effectively embed identification signals. ***Antidistillation fingerprinting*** (ADFP) is a novel fingerprinting scheme that aligns the goal of distillation detection via watermarking with the student's natural learning dynamics. Specifically, ADFP utilizes a proxy model to identify and sample tokens that approximately maximize the student's expected probability of generating green-list tokens after fine-tuning. By dynamically targeting tokens that efficiently drive the student's optimization trajectory toward the fingerprint, ADFP achieves a significantly improved trade-off between fingerprinting effect and generation quality.

We validate this advantage on GSM8K, OASST1, and MBPP, spanning mathematical reasoning, open-domain dialogue, and code generation. Across all three domains, ADFP

---

[1]Carnegie Mellon University, Pittsburgh, PA, USA [2]University of Maryland, College Park, MD, USA [†]Asher is now at Google. Correspondence to: Yixuan Even Xu <yixuanx@cs.cmu.edu>.

*Proceedings of the $43^{rd}$ International Conference on Machine Learning*, Seoul, South Korea. PMLR 306, 2026. Copyright 2026 by the author(s).

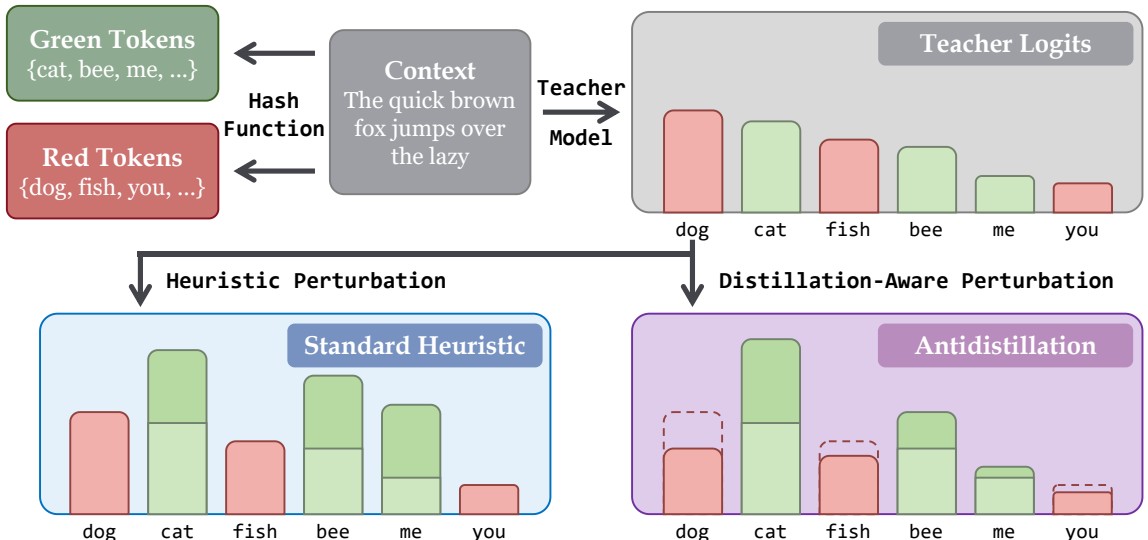

*Figure 1.* Antidistillation fingerprinting (ADFP) performs targeted logit perturbations aligned with the student's learning dynamics to optimize fingerprinting effect. Visually, while the standard heuristic boosts green tokens uniformly, ADFP selectively amplifies high-likelihood ones, which are most likely to be internalized, improving the quality-fingerprinting trade-off.

achieves a significant Pareto improvement over the state-of-the-art red-and-green-list baseline. As an example, at a fixed level of generation quality measured by teacher accuracy on GSM8K, ADFP lowers the expected false positive rate by nearly an order of magnitude ($p$-value 0.09 versus 0.01). Our results indicate that model owners can leverage ADFP to reliably detect distillation of their models.

## 2. Related Work

**MIA and Memorization.** A membership inference attack (MIA, Shokri et al. (2017)) is a game where an attacker tries to determine whether or not a specific sample, or set of samples (Maini et al., 2021), were included in a model's training dataset. Multiple studies surveying state of the art techniques have shown that in modern settings like LLM pretraining, most methods achieve MIA success close to random chance (Duan et al., 2024; Hayes et al., 2025).

Independently, a broad array of work has shown that pretraining LLMs at web scale causes them to memorize parts of their training data verbatim, typically proportional to a sample's frequency (Carlini et al., 2021; 2022). However, while memorization checks may seem like an intuitive candidate for a membership inference criterion, memorization and regurgitation risk is not uniform across a training corpus, and appears to be minimal in practice relative to corpus size (Cooper et al., 2025). Therefore, using the reproduction of a partial sample as a decision criterion is insufficient for reliable membership inference. Models may fail to regenerate samples they were indeed trained on, while simultaneously happening to "generalize" and produce sequences

they were never actually trained on (Liu et al., 2025).

**Distillation.** Distillation of neural networks is a well established set of techniques for transferring knowledge and behavior from a teacher model into a student model. Ten years of literature on distillation exists (Hinton et al., 2015; Gou et al., 2021) and the modern foundation model paradigm heavily leverages both direct logit distillation and indirect techniques like training on model-generated datasets in the latter phases of multi-stage training pipelines (see Xu et al. (2024) for a comprehensive survey). There is also a fundamental connection between distillation and MIA (Jagielski et al., 2024) as distillation success and MIA success are both predicated on the model absorbing measurable, and potentially inferrable, information from its training data. Our research is particularly motivated by this theoretical connection combined with claims that current industry model providers may be distilling from competitors in violation of their terms of service (Singh, 2025). Currently, parties lack statistically grounded methods to evidence such assertions.

**Output Watermarks.** Recent research has explored numerous variations of *output* watermarks for LLMs, analyzing the inherent tradeoffs between their robustness and stealthiness in practice (Aaronson, 2022; Kirchenbauer et al., 2023; Fairoze et al., 2023; Fernandez et al., 2023; Christ et al., 2024; Kuditipudi et al., 2024; Dathathri et al., 2024; Piet et al., 2025; Ma et al., 2026). Cross-cutting research from academia and industry has demonstrated that watermarks render the teacher's outputs "radioactive," thereby imprinting a detectable signature onto any student model trained on them (Sander et al., 2024). This suggests the feasibility of threat models like watermark spoofing (Gu et al., 2023) and

new applications like contamination detection via proactive planting of marked samples in publicly released benchmark datasets (Sander et al., 2025). Our work builds upon these key results by leveraging the specific technique of antidistillation sampling (Savani et al., 2025) to show that in the presence of watermarks, distillation detection becomes a highly tractable instance of membership inference.

## 3. Formulations

**Basic setting.** We consider the scenario where a *student model* $\theta_s$ is fine-tuned on the outputs of a *teacher model* $\theta_t$. The owner of the teacher model wants to fingerprint the outputs so that later they can detect whether a student model is fine-tuned on the teacher's outputs. The owner does not know the architecture or weights of the student model in advance. Therefore, the fingerprinting scheme can only rely on a *proxy student model* $\theta_p$. The models are autoregressive language models with vocabulary $\mathcal{V}_t, \mathcal{V}_s, \mathcal{V}_p$, respectively.

For model $\theta \in \{\theta_t, \theta_s, \theta_p\}$, we denote its logits and predicted distribution at step $l$ given context $x_{1:l}$ as $z(\cdot \mid x_{1:l}; \theta)$ and $q(\cdot \mid x_{1:l}; \theta)$, respectively. We use $T$ to denote a tokenizer for the models, where $T(s, \mathcal{V})$ maps a string $s$ to a token sequence in vocabulary $\mathcal{V} \in \{\mathcal{V}_t, \mathcal{V}_p, \mathcal{V}_s\}$.

**Red-and-green-list watermarking.** Our method builds upon the red-and-green-list watermarking scheme proposed by Kirchenbauer et al. (2023). In this scheme, there is a predetermined hash function $H : \mathcal{V}_t^w \times \mathcal{K} \to 2^{\mathcal{V}_t}$, where $w$ is the hash window size and $\mathcal{K}$ is the key space. The hash function returns a *green list* of $\gamma \cdot |\mathcal{V}_t|$ tokens. Here, $\gamma \in (0, 1)$ is the green-list ratio. In this work, we assume independent hash values across different inputs.

The watermarking is done by perturbing the teacher model's logits. At step $l$, given context $x_{1:l}$, a **red-and-green-list logit perturbation** $\Delta_t^{\mathrm{RGL}}$ is computed, where

$$\Delta_t^{\mathrm{RGL}} = \mathbb{I}\Big[t \in H(x_{-w:}, k)\Big]. \qquad (1)$$

Here, $x_{-w:}$ denotes the last $w$ tokens of $x$. Let $z = z(\cdot \mid x_{1:l}; \theta_t)$ be the teacher model's logits at step $l$. The logits are perturbed with $\tilde{z} = z + \delta \cdot \Delta^{\mathrm{RGL}}$ for parameter $\delta > 0$.

For an unwatermarked piece of text, the fraction of green-list tokens sampled is expected to concentrate around $\gamma$, while for a watermarked one, the fraction should be significantly higher. Thus, to detect the watermark, a statistical test on the fraction of green-list tokens can be performed.

**Statistical fingerprinting.** Sander et al. (2024; 2025) have shown that training, or distilling a model from the outputs of another model known to have been deployed with the red-and-green-list watermark imprints the downstream model with a diluted version of the same watermark. Their results demonstrate the basic transfer effect necessary for an output

watermark to be treated as a "fingerprint" useful for determining whether distillation was performed. In this section we formalize this threat model and derive a more principled approach for robust anti-distillation fingerprinting with red-and-green-list watermarks.

In the fingerprinting context, the detection is based on the statistics which we name as **average green-list token probability (GTP)**. Given an evaluation dataset $\mathcal{X} = \{x_1, \ldots, x_n\}$, where each $x_i$ is a token sequence longer than $w$ and the last $w$ tokens of $x_i$ are distinct for all $i$, $\mathrm{GTP}(\mathcal{X}, \theta_s, k)$ is defined as the average probability that the student model $\theta_s$ generates a green-list token after each context $x_i$ using the key $k$ to compute the green list.

Depending on whether the student model $\theta_s$ is open-weight or closed-weight, the estimation of $\mathrm{GTP}(\mathcal{X}, \theta_s, k)$ is done differently. If the student model is open-weight, we can compute $\mathrm{GTP}(\mathcal{X}, \theta_s, k)$ directly from the model logits, i.e.,

$$\frac{1}{n} \sum_{i=1}^n \mathbf{Pr}\left[\text{Next token } t \text{ after } x_i \text{ by } \theta_s \in H(x_{i,-w:}, k)\right].$$

Otherwise, for closed-weight models, we define it as a random variable by sampling next tokens for each context in $\mathcal{X}$ once and computing the fraction of green-list tokens, i.e.,

$$\frac{1}{n} \sum_{i=1}^n \mathbb{I}\Big[\text{Sampled token } t \text{ after } x_i \text{ by } \theta_s \in H(x_{i,-w:}, k)\Big].$$

The fingerprint detection is done by a hypothesis test.

**Null hypothesis.** *The generation of the student model $\theta_s$ is independent of the key $k$ used to fingerprint the teacher.*

**Lemma 3.1.** *Under the null hypothesis and the randomness of $k$, for both the open-weight and closed-weight evaluation, $\mathrm{GTP}(\mathcal{X}, \theta_s, k)$ is a mean of $n$ independent random variables, each bounded in $[0, 1]$ with mean $\gamma$.*

We provide the proof of Lemma 3.1 in Section A.1. Lemma 3.1, shows that under the null hypothesis, $\mathrm{GTP}(\mathcal{X}, \theta_s, k)$ is a mean of bounded independent random variables with mean $\gamma$. Therefore, $\mathrm{GTP}(\mathcal{X}, \theta_s, k)$ concentrates around $\gamma$. Using Hoeffding's inequality, we can derive a conservative $p$-value for the fingerprint detection.

**Theorem 3.1.** *Let $g_{\mathrm{obs}} > \gamma$ be an observed empirical value $\mathrm{GTP}(\mathcal{X}, \theta_s, k)$. Under the null hypothesis, a conservative $p$-value is $p = \exp\big(-2n(g_{\mathrm{obs}} - \gamma)^2\big)$.*

## 4. Methodology

### 4.1. Antidistillation Logit Perturbation

Kirchenbauer et al. (2023) used a simple logit perturbation $\tilde{z} = z + \delta \cdot \Delta^{\mathrm{Base}}$ to encourage sampling from the green list

---

**Algorithm 1:** Antidistillation Fingerprinted Sampling

**Input:** Teacher and proxy model $\theta_t, \theta_p$, Hash function $H$, Hash key $k$, tokenized prompt $x_{1:l}$, penalty multiplier $\lambda$, temperature $\tau$, window size $w$

(1). Green list at the step $S \leftarrow H(x_{-w:}, k)$

(2). Compute the proxy's prediction $q(\cdot | x_{1:l}; \theta_p)$

(3). Compute the logit perturbation $\Delta^{\text{ADS}}$ using (2)

(4). Sample the next token $x_{l+1}$

$$x_{l+1} \sim \frac{1}{Z} \exp\left( \frac{1}{\tau} \log q(\cdot | x_{1:l}; \theta_t) + \lambda \Delta^{\text{ADS}} \right)$$

**Output:** Sampled next token $x_{l+1}$

---

**Algorithm 2:** Antidistillation Fingerprint Detection

**Input:** Student model $\theta_s$, Hash function $H$, Hash key $k$, evaluation dataset $\mathcal{X}$, window size $w$

(1). Compute $g_{\text{obs}} \leftarrow \text{GTP}(\mathcal{X}, \theta_s, k)$

(2). Return $p$-value $p \leftarrow \exp\left( -2n(g_{\text{obs}} - \gamma)^2 \right)$

**Output:** A $p$-value $p$ for the fingerprint detection

---

at each step for parameter $\delta > 0$. Let the green list at step $l$ be $S \equiv S(x_{1:l})$. Their perturbation is $\Delta_t^{\text{Base}} = \mathbb{I}[t \in S]$.

Using the idea of antidistillation sampling (ADS) (Savani et al., 2025), we can derive a more principled logit perturbation, $\Delta^{\text{ADS}}$, that directly optimizes the expected green-list probability used in statistical fingerprinting on a proxy model $\theta_p$. Let $q \equiv q(\cdot \mid x_{1:l}; \theta_p)$ be the proxy model's predicted distribution at step $l$. $\Delta^{\text{ADS}}$ is given by

$$\Delta_t^{\text{ADS}} = q_t \cdot \left( \mathbb{I}[t \in S] - L \right), \text{where } L = \sum_{t \in S} q_t. \quad (2)$$

Intuitively, $\Delta_t^{\text{ADS}}$ focuses more on tokens that the proxy model believes are likely to be sampled, i.e., with large $q_t$. The factor $\mathbb{I}[t \in S] - L$ is an advantage baseline: it encourages sampling from the green list (when $\mathbb{I}[t \in S] = 1$) and discourages sampling from the red list (when $\mathbb{I}[t \in S] = 0$).

We define the **antidistillation logit perturbation** as $\tilde{z} = z + \lambda \cdot \Delta^{\text{ADS}}$ for parameter $\lambda > 0$, where $\Delta^{\text{ADS}}$ is given by (2). In Section 4.2, we provide a detailed derivation of (2).

### 4.2. Derivation of Antidistillation Logit Perturbation

Consider at context $x_{1:l}$, we are about to generate the next token. Intuitively, for a token $t$, if after fine-tuning the proxy model on $t$, its expected green-list probability increases, we want to increase the likelihood of sampling $t$ at this step.

**Per step loss.** Specifically, let $z \equiv z(\cdot \mid x_{1:l}; \theta_p)$ be the

logits of the proxy model $\theta_p$ at context $x_{1:l}$, and $q$ be the corresponding softmax distribution. Therefore,

$$q_t \equiv \text{softmax}(z)_t = \frac{e^{z_t}}{\sum_{i \in \mathcal{V}_p} e^{z_i}}.$$

Given a green list $S \equiv S(x_{1:l})$, define the **per-step loss**

$$L \equiv L(x_{1:l}) = \sum_{t \in S} q_t = \mathbf{E}_{t \sim q}\left[ \mathbb{I}[t \in S] \right]. \quad (3)$$

**Antidistillation sampling.** We generalize antidistillation sampling (ADS) (Savani et al., 2025) to our setting. In ADS, we likewise want to increase the likelihood of sampling tokens that increase a *downstream loss* after fine-tuning. The downstream loss $L$ is a function of $\theta_p$, *independent* of the current context $x_{1:l}$. According to equation (8) in (Savani et al., 2025), the desired logit perturbation $\Delta$ is

$$\Delta_t = \left\langle \nabla_{\theta_p} \log q_t, \nabla_{\theta_p} L \right\rangle. \quad (4)$$

In our case, the loss $L$ is context-dependent, i.e., the *per step loss* defined in (3). Thus, directly computing $\Delta$ using (4) is expensive, as it requires a backward pass for each token $t$ in the vocabulary. In the following, we derive an approximation of $\Delta$ that admits an efficient computation.

**Softmax Jacobian.** Since $q = \text{softmax}(z)$, the Jacobian of the probabilities $q$ with respect to logits $z$ is

$$\frac{\partial q_t}{\partial z_i} = q_t \left( \mathbb{I}[i = t] - q_i \right).$$

**Gradient of $L$ in logit space.** Recall $L = \sum_{t \in S} q_t$, thus

$$\frac{\partial L}{\partial z_i} = \sum_{t \in S} \frac{\partial q_t}{\partial z_i} = \sum_{t \in S} q_t \left( \mathbb{I}[i = t] - q_i \right)$$
$$= q_i \cdot \mathbb{I}[i \in S] - q_i \sum_{t \in S} q_t = q_i \left( \mathbb{I}[i \in S] - L \right).$$

Denote the logit-space gradient vector as $r$, i.e.,

$$r_i \equiv \frac{\partial L}{\partial z_i} = q_i \left( \mathbb{I}[i \in S] - L \right).$$

Note that $\sum_i r_i = \sum_i q_i(\mathbb{I}[i \in S] - L) = L - L = 0$.

**From logit space to parameter space.** Let $J \equiv \frac{\partial z}{\partial \theta_p}$ be the Jacobian of the logits with respect to the model parameters, and $e_t$ be the one-hot vector for token $t$. Then

$$\nabla_{\theta_p} \log q_t = J^\top (e_t - q), \text{ and } \nabla_{\theta_p} L = J^\top r.$$

Hence the inner product defined in (4) is

$$\Delta_t = \left\langle \nabla_{\theta_p} \log q_t, \nabla_{\theta_p} L \right\rangle = (e_t - q)^\top \underbrace{(JJ^\top)}_{K} r, \quad (5)$$

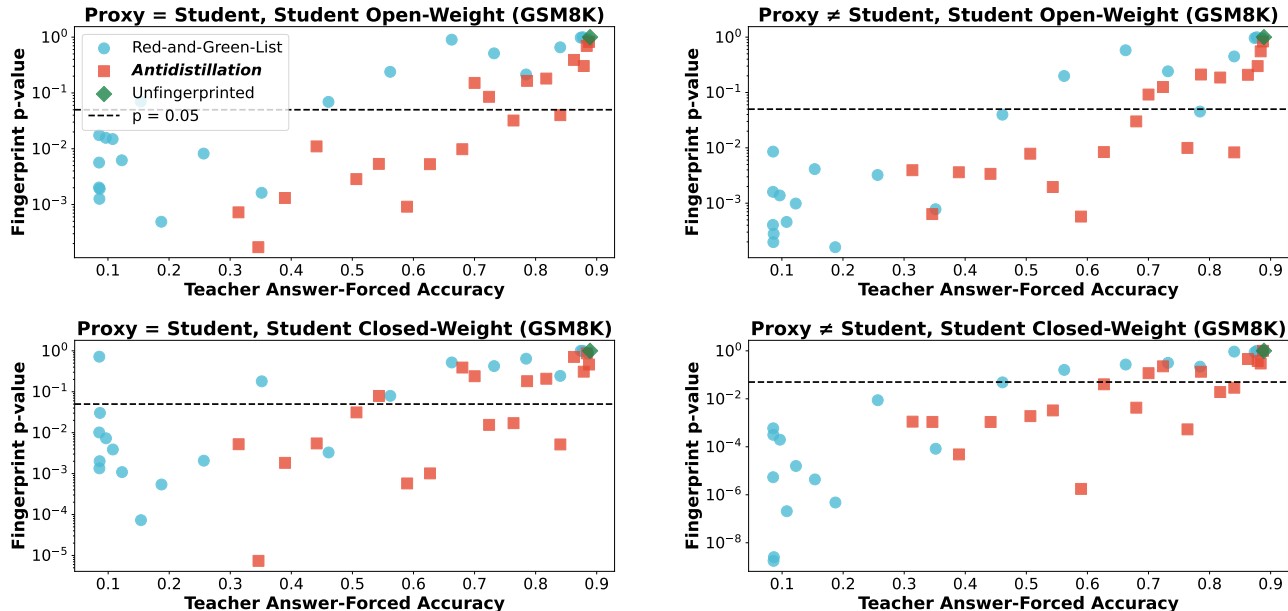

*Figure 2.* Trade-off between fingerprinting $p$-value and generation quality on GSM8K under unsupervised evaluation. Each point corresponds to a different logit perturbation strength $\delta$ or $\lambda$. Lower $p$-value indicates stronger fingerprinting effect, and higher accuracy indicates better generation quality. Antidistillation fingerprinting achieves a pareto improvement over red-and-green-list fingerprinting.

where $K \equiv JJ^\top$ is a positive semidefinite matrix.

**Isotropic approximation.** Note that if we let $g_i = \nabla_{\theta_p} z_i$, then $K$ can be expressed as $K_{i,j} = \langle g_i, g_j \rangle$, i.e., the Gram matrix of the gradients of the logits with respect to the model parameters. Since directly computing (5) is expensive, we propose to make the isotropic approximation that $K \approx c \cdot I$ for some constant $c > 0$. The intuition is that, most directions in the logit space should be approximately equally important. We also prove that the assumption holds exactly for the following special case to further justify it.

**Proposition 4.1.** *If the proxy model $\theta_p$ only has a trainable final linear adaptation layer, then $K \propto I$ holds exactly.*

The proof of Proposition 4.1 is provided in Section A.2.

Using the approximation, (5) yields

$$\Delta_t \propto (e_t - q)^\top r \ = \ r_t - q^\top r.$$

For these two terms, we have

$$r_t = q_t \left( \mathbb{I}\Big[t \in S\Big] - L \right),$$

$$q^\top r = \sum_{i \in \mathcal{V}_p} q_i \cdot q_i \left( \mathbb{I}\Big[i \in S\Big] - L \right) = \sum_{i \in S} q_i^2 - L \sum_{i \in \mathcal{V}_p} q_i^2.$$

The quantity $q^\top r$ does *not* depend on the specific token $t$ being considered. Therefore, it is a constant shift that cancels under the transformations used for sampling (e.g., linear re-normalizing). Thus, we may drop it and define the

**antidistillation logit perturbation score** as

$$\boxed{\Delta_t^{\mathrm{ADS}} \equiv r_t = q_t \left( \mathbb{I}\Big[t \in S\Big] - L \right), \text{ where } L = \sum_{t \in S} q_t.}$$

Note that because $\Delta^{\mathrm{ADS}}$ is derived from the gradient of the per-step loss $L$, it directly optimizes the expected green-list token probability on the proxy model. Therefore, intuitively, using $\Delta^{\mathrm{ADS}}$ for logit perturbation is expected to yield better fingerprinting performance than using the simple red-and-green-list perturbation, or other heuristic perturbations.

### 4.3. Antidistillation Fingerprinting

**Definition 4.1.** *Antidistillation Fingerprinting (ADFP) is a fingerprinting scheme consisting of a sampling algorithm and a detection algorithm with a shared secret key. The algorithms are given in Algorithms 1 and 2, respectively.*

To apply ADFP, the teacher model owner first selects a hash function $H$ and generates a secret key $k$. When responding to generation requests, the teacher model uses Algorithm 1 to sample the next token at each step. To evaluate whether a student model $\theta_s$ is fine-tuned on the teacher's outputs, the model owner uses the secret ket $k$ and Algorithm 2 to compute a $p$-value for the fingerprint detection.

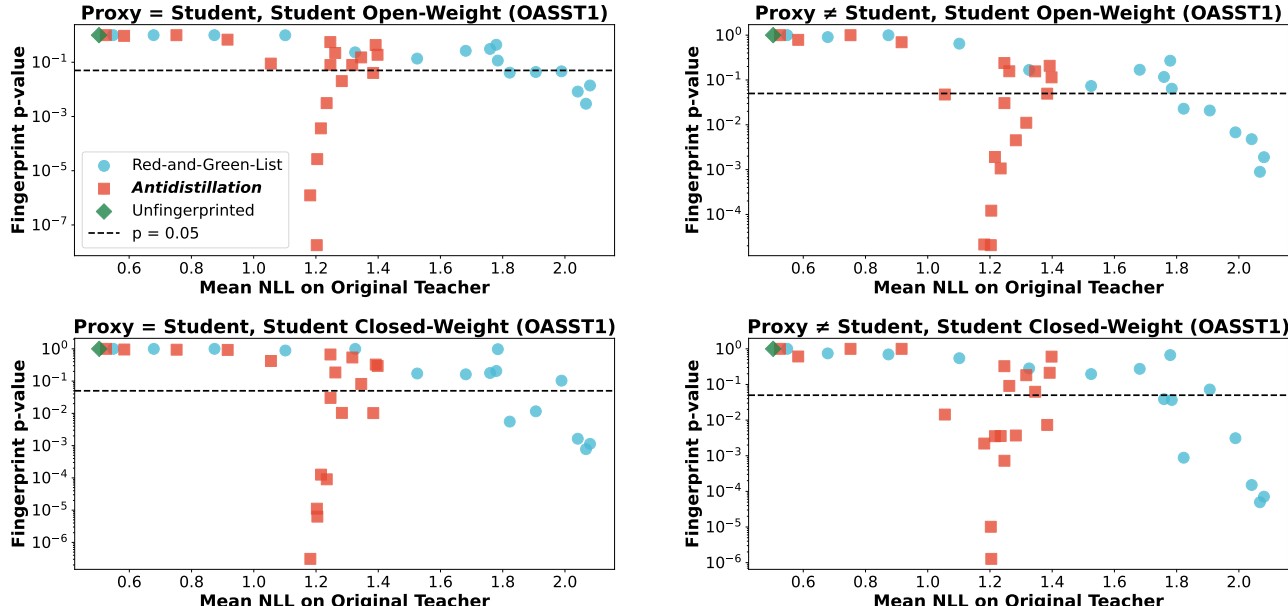

*Figure 3.* Trade-off between fingerprinting $p$-value and generation quality on OASST1 under unsupervised evaluation. Each point corresponds to a different logit perturbation strength $\delta$ or $\lambda$. Lower $p$-value indicates stronger fingerprinting effect, and lower NLL indicates better generation quality. Antidistillation fingerprinting achieves a pareto improvement over red-and-green-list fingerprinting.

## 5. Experiments

### 5.1. Experimental Setup

We conduct experiments to evaluate the effectiveness of ADFP in fingerprinting student models fine-tuned on the teacher's outputs. We first describe the experimental setup.

**Algorithms.** We compare ADFP with the red-and-green-list fingerprinting (Kirchenbauer et al., 2023; Sander et al., 2024) described in Section 3. The methods are denoted as *antidistillation* and *red-and-green-list* in the experiments, respectively. Both methods use the same hash function, key and fingerprint detection process for fair comparison.

**Models.** To simulate a realistic distillation scenario, we use distinct teacher and proxy models. We consider two scenarios where the student is the same as or different from the proxy. For GSM8K and OASST1, we use `deepseek-ai/DeepSeek-R1-Distill-Qwen-7B` (Guo et al., 2025) as the teacher model, `Qwen/Qwen2.5-3B` (Team et al., 2024) as the proxy model, and either `Qwen/Qwen2.5-3B` or `meta-llama/Llama-3.2-3B` (Grattafiori et al., 2024) as the student model. For MBPP, we use the code-specialized `Qwen/Qwen2.5-Coder-7B-Instruct` teacher together with `Qwen/Qwen2.5-Coder-3B-Instruct` as the proxy, and either `Qwen/Qwen2.5-Coder-3B-Instruct` or `meta-llama/Llama-3.2-3B-Instruct` as the student model.

**Benchmarks.** We consider three domains for distillation: mathematical reasoning, open-domain dialogue, and code generation. For mathematical reasoning, we use the GSM8K dataset (Cobbe et al., 2021), which consists of grade school math problems. We use all 7473 problems in the training set, and construct prompts by concatenating each problem with a system prompt in the model's chat template. For open-domain dialogue, we use the OASST1 dataset (OpenAssistant Contributors, 2023), which contains multi-turn conversations. We construct a subsampled dataset of 8192 prompts, each by concatenating the conversation history until an assistant message with a system prompt in the model's chat template. For code generation, we use MBPP (Austin et al., 2021). We combine the official train, validation, and test pools, use the first 70% (674 problems) for fingerprinted generation and student fine-tuning. Following a standard function-completion setup, each prompt includes the problem description and one public assertion, and teacher utility is measured by execution pass rate and syntax validity.

**Experimental pipeline.** Throughout the experiments, we set the hash window size $w = 2$ and green-list fraction $\gamma = 0.5$. The experimental pipeline consists of 3 stages.

- *Stage 1: Fingerprinted generation.* We use the teacher model to generate watermarked outputs for the prompts using either ADFP or red-and-green-list fingerprinting.

- *Stage 2: Student fine-tuning.* We fine-tune the student model on the generated outputs using LoRA (Hu et al., 2022) with rank 128 and $\alpha = 128$. We use 1 epoch for GSM8K and OASST1, and 3 epochs for MBPP because its training split is substantially smaller.

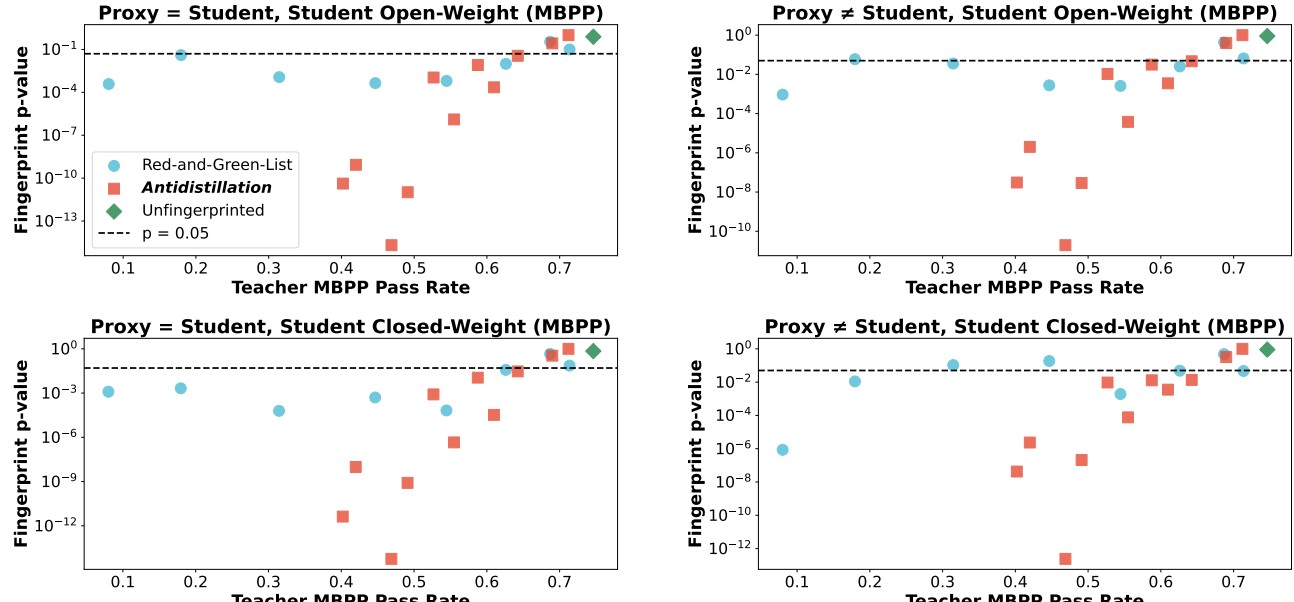

*Figure 4.* Trade-off between fingerprinting $p$-value and generation quality on MBPP under unsupervised evaluation. Each point corresponds to a different logit perturbation strength $\delta$ or $\lambda$. Lower $p$-value indicates stronger fingerprinting effect, and higher execution pass rate indicates better generation quality. Antidistillation fingerprinting achieves a pareto improvement over red-and-green-list fingerprinting.

- *Stage 3: Evaluation.* We evaluate both the quality of the generated outputs by the teacher model and the fingerprinting effect on the fine-tuned student model.

We provide the hyperparameter details in Section A.3.

**Evaluation.** From the teacher model owner's perspective, it is not clear what fine-tuning data the student model uses. Therefore, we mainly consider the *unsupervised* evaluation setting mentioned in Sander et al. (2024), where the evaluation dataset $\mathcal{X}$ is related to, but distinct from, the student model's fine-tuning data. In this work, for GSM8K and OASST1, we generate $\mathcal{X}$ by generating outputs for the same prompts used in *Stage 1* with a different random seed, while for MBPP we generate $\mathcal{X}$ from the same prompts using alternative decoding settings. In Section B.1, we also consider the *supervised* evaluation setting, where the evaluation dataset $\mathcal{X}$ is exactly the student model's fine-tuning data.[1]

We consider both open-weight and closed-weight student models in the evaluation. For open-weight models, we compute the average green-list token probability $\mathrm{GTP}(\mathcal{X}, \theta_s, k)$ directly from the model logits. For closed-weight models, we sample next tokens for each context in $\mathcal{X}$ once and

compute the fraction of green-list tokens. As we will see later, both evaluation settings yield similar fingerprinting $p$-values. However, intuitively, open-weight evaluation should be more powerful. This is because closed-weight evaluation consumes more resources in real applications to achieve the same statistical power, as detailed in Section A.4.

For both algorithms and all experimental settings, we vary the fingerprinting strength, i.e., $\lambda$ for ADFP and $\delta$ for red-and-green-list fingerprinting, to obtain different trade-offs between generation quality and fingerprinting effect. We measure the fingerprinting effect using the $p$-value defined in Theorem 3.1. A smaller $p$-value indicates a stronger effect. For generation quality, we use *answer-forced accuracy* for GSM8K, *NLL on original teacher* for OASST1, and *execution pass rate* together with *syntax validity* for MBPP. Here, answer-forced accuracy is computed by concatenating `\n\n**FinalAnswer**\n[\boxed{` after the reasoning trace and continue to generate for 32 additional tokens and checking if the final answer is correct. NLL on original teacher is computed by evaluating the negative log-likelihood of the traces with the unfingerprinted teacher.

We provide additional evaluation details in Section A.4.

**Implementation.** We run our experiments on nodes with 8 NVIDIA H100 GPUs or 8 AMD MI250X GPUs each, and we implement the code with the `transformers` (Wolf et al., 2020), `trl` (von Werra et al., 2020), and `accelerate` (Gugger et al., 2022) libraries. Student accuracy measurements on the GSM8K test set are per-

---

[1]In the MIA literature, two sets are often considered: "member" samples that the target model actually trained on and "non-member" samples that are held out and known to not have been trained on. In our work, we consider "whether the student has trained on *any* data samples with this fingerprint" rather than asking about training on specific instances. Hence, we choose not to directly adopt member / non-member terminology.

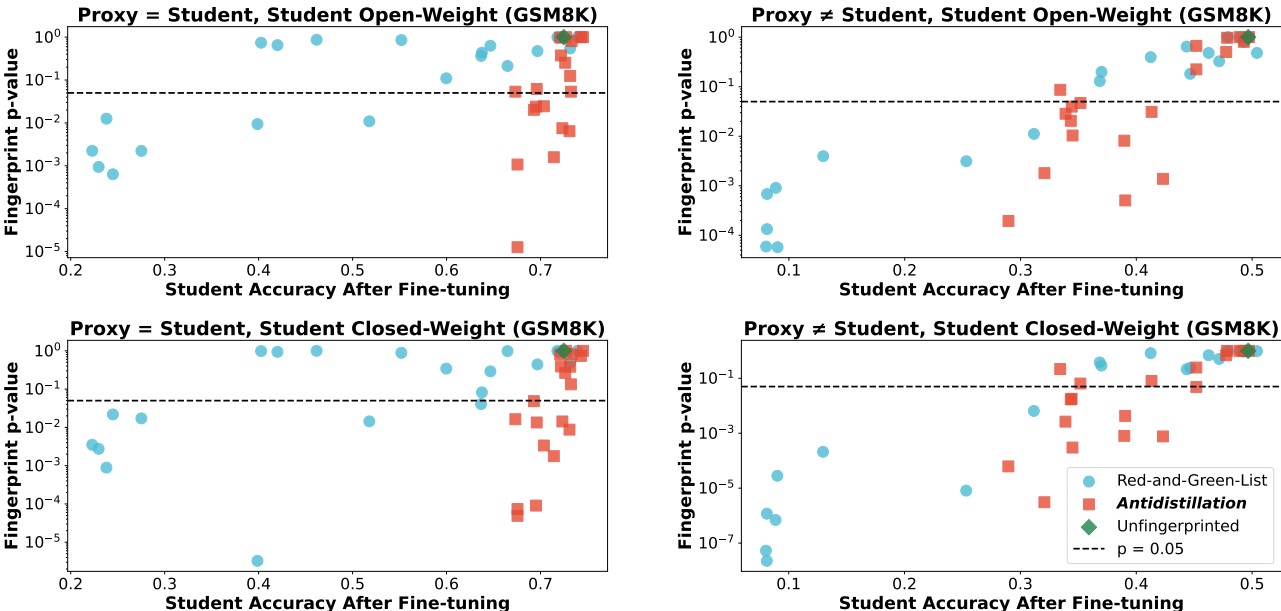

*Figure 5.* Trade-off between fingerprinting $p$-value and student's accuracy after fine-tuning on GSM8K. Each point corresponds to a different logit perturbation strength $\delta$ or $\lambda$. Lower $p$-value indicates stronger fingerprinting effect, and higher accuracy indicates better fine-tuning quality. Antidistillation fingerprinting achieves a pareto improvement over red-and-green-list fingerprinting.

formed using the implementation and standard settings provided by the Eleuther LM Evaluation Harness (Gao et al., 2024). Our code for running the experiments is released at `https://github.com/YixuanEvenXu/antidistillation-fingerprinting`.

### 5.2. Experimental Results

**Fingerprinting effect.** We present the trade-off between fingerprinting $p$-value and generation quality under unsupervised evaluation in Figures 2 to 4. We observe that ADFP consistently achieves a pareto improvement over red-and-green-list fingerprinting across the experimental settings we test. The results concur with our theoretical analysis that ADFP directly optimizes the expected green-list token probability on the proxy model, leading to a stronger fingerprinting effect for the same quality degradation, which demonstrates the effectiveness of ADFP in fingerprinting student models fine-tuned on the teacher's outputs.

**Fine-tuning quality.** We then present the trade-off between fingerprinting $p$-value and student's accuracy after fine-tuning on GSM8K in Figure 5. In this set of experiments, the p-values are computed over the teacher generated evaluation dataset $\mathcal{X}$ in the same manner as Figure 2 but the "student accuracy" is independently measured for each trained student model on the official GSM8K test set. We observe that ADFP also achieves a pareto improvement over red-and-green-list fingerprinting across the experimental settings. This shows that the output texts generated by ADFP

are less likely to harm the fine-tuning quality of the student model while secretly embedding a fingerprint, making ADFP less detectable in practice. Notably, for settings where the proxy model is the same as the student model, ADFP causes only negligible fine-tuning quality degradation, even with strong fingerprinting effect, in contrast to red-and-green-list fingerprinting.

**Partially fingerprinted fine-tuning data.** In realistic scenarios, frontier models do not wish to sacrifice generation quality for common use cases, and the logit perturbation is only applied when a stream of requests are suspected to be used for model extraction. As a result, student models may be fine-tuned on a mixture of fingerprinted and non-fingerprinted data. We consider when the student model is fine-tuned on a mixture of $\alpha\%$ fingerprinted outputs generated by the teacher using ADFP or red-and-green-list fingerprinting, and $(100 - \alpha)\%$ unfingerprinted outputs generated by the teacher without logit perturbation.

Specifically, we pick $\lambda = 256$ for ADFP and $\delta = 7$ for red-and-green-list fingerprinting, which yield similar teacher accuracies (52% for ADFP and 47% for red-and-green-list fingerprinting) on GSM8K. We vary $\alpha$ from 0 to 100 and present the fingerprinting $p$-value for each kind of evaluation in Figure 6. We observe that as $\alpha$ decreases, the fingerprinting effect of both methods degrades. However, ADFP remains more effective than red-and-green-list fingerprinting across most values of $\alpha$ given a similar generation quality, except when $\alpha$ is too small that the fingerprinting effect is weak for both methods. This demonstrates the ap-

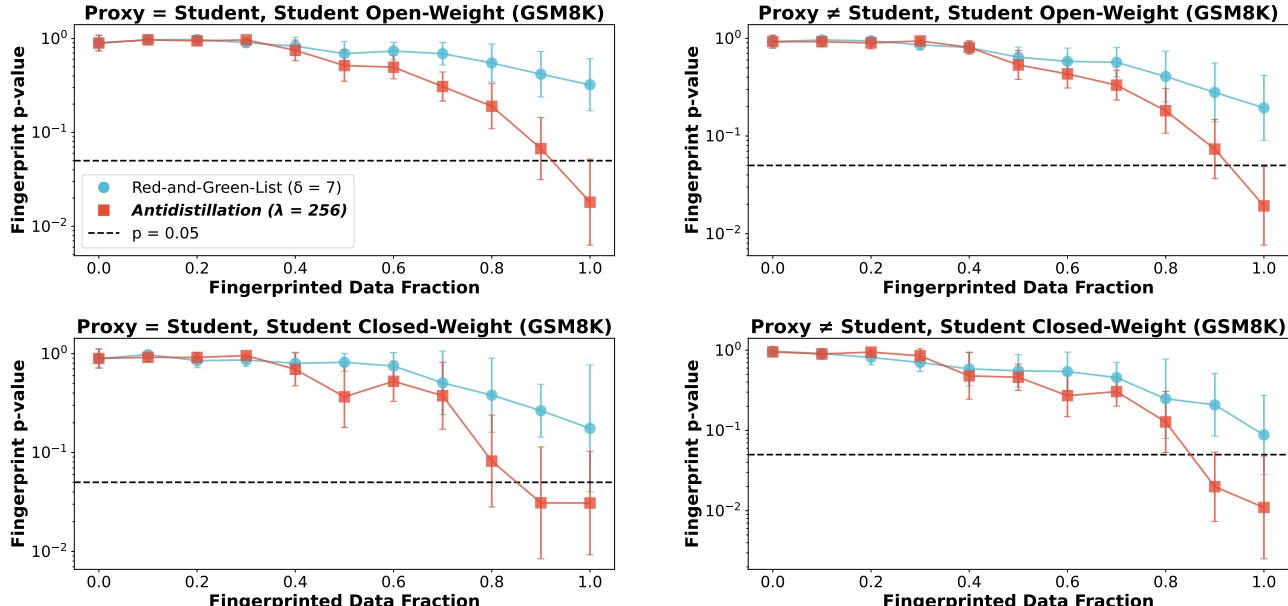

*Figure 6.* The effect of fingerprinted data fraction on fingerprinting $p$-value for both antidistillation fingerprinting ($\lambda = 256$, teacher accuracy 52%) and red-and-green-list fingerprinting ($\delta = 7$, teacher accuracy 47%) on GSM8K. Each data point is averaged over 10 random trials in log space, with error bars indicating 1.96 times standard error of mean. Both methods' fingerprinting effect degrades as the fingerprinted data fraction decreases, but antidistillation fingerprinting remains more effective.

plicability of ADFP in practical scenarios where the student model may be fine-tuned on partially fingerprinted data.

**Additional experiments.** In Section B, we provide additional experimental results, including results under supervised evaluation (Section B.1), alternative student fine-tuning settings (Section B.2), an empirical study of true and false positive rates (Section B.3), and qualitative examples of generated outputs (Section B.4).

## 6. Conclusion

In this paper, we introduced antidistillation fingerprinting (ADFP), a principled framework for detecting model distillation. Our experiments on GSM8K, OASST1, and MBPP demonstrate that ADFP achieves a significant Pareto improvement over state-of-the-art baselines across mathematical reasoning, dialogue, and code generation. Theoretically, by directly optimizing the detection objective within the student's learning dynamics, ADFP maximizes the fingerprint's learnability. This formulation not only justifies ADFP's superior performance but also offers a theoretical lens to understand why previous heuristic watermarks persist during fine-tuning. Qualitatively, we observe that ADFP generates more coherent and less repetitive text than baselines at comparable fingerprinting strengths. Consequently, ADFP provides model owners with a potent and statistically grounded tool for intellectual property protection.

## Acknowledgments

Researcher YEX was supported in part by NSF grant IIS-2046640 (CAREER). YEX would like to acknowledge the helpful discussion with Zhili Feng at OpenAI on topics relating to antidistillation sampling and watermarking.

Researcher JK was supported by DARPA TIAMAT, the NSF TRAILS Institute (2229885), and Coefficient Giving. Computing resources for this project were supported in part by the U.S. Department of Energy, Office of Science, Office of Advanced Scientific Computing Research under the Advancements in AI for Science program. JK would like to acknowledge fruitful discussions with Pierre Fernandez and Tom Sander at Meta on topics relating to the intersection of watermarking, membership, and attribution.

## Impact Statement

This work contributes to the robust identification of large language model provenance, addressing the growing tension between open model release and intellectual property protection. As pre-training frontier models requires substantial resources, the ability to reliably detect third-party distillation is essential for the sustainability of the foundation model ecosystem. ADFP advances this goal by providing a statistically grounded mechanism that not only effectively detects infringement but also, crucially, minimizes the risk of false accusations through rigorous $p$-value estimation. By offering model owners a trustworthy tool to enforce license

terms, our method may incentivize the continued public release and API access of high-performance models.

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

# A. Missing Details

## A.1. Proof of Lemma 3.1

**Lemma 3.1.** *Under the null hypothesis and the randomness of $k$, for both the open-weight and closed-weight evaluation,* $\mathrm{GTP}(\mathcal{X}, \theta_s, k)$ *is a mean of $n$ independent random variables, each bounded in $[0, 1]$ with mean $\gamma$.*

**Proof of Lemma 3.1:** Since the contexts $x_i$ are distinct in their last $w$ tokens, by the definition of a hash function, for any hash key $k$, the green lists $H(x_{i,-w:}, k)$ are independent and uniformly random subsets of the vocabulary with their sizes being $\gamma$ fraction of the vocabulary size. Under the null hypothesis, the generation of the student model $\theta_s$ is independent of the key $k$. Thus for each $i$, the generation distribution of the student model after context $x_i$ is independent of the randomness of the green list $H(x_{i,-w:}, k)$. Consider the following two cases:

- For open-weight evaluation, this implies that for each $i$, the probability $\mathbf{Pr}\left[\text{Next token } t \text{ after } x_i \text{ by } \theta_s \in H(x_{i,-w:}, k)\right]$ is independent of the randomness of $H(x_{i,-w:}, k)$. Since for each $i$, $H(x_{i,-w:}, k)$ is a uniformly random subset of fraction $\gamma$, we know that these probabilities are independent random variables with mean $\gamma$.

- For closed-weight evaluation, this implies that for each $i$, the sampled token $t$ after $x_i$ by the student model $\theta_s$ is independent of the randomness of the green list $H(x_{i,-w:}, k)$. We similarly know that the indicators $\mathbb{I}\left[\text{Sampled token } t \text{ after } x_i \text{ by } \theta_s \in H(x_{i,-w:}, k)\right]$ are independent random variables with mean $\gamma$.

This concludes the proof of Lemma 3.1. ∎

## A.2. Proof of Proposition 4.1

**Proposition 4.1.** *If the proxy model $\theta_p$ only has a trainable final linear adaptation layer, then $K \propto I$ holds exactly.*

**Proof of Proposition 4.1:** Suppose that for context $x_{1:l}$, the activation vector in the proxy model before the final linear adaptation layer is $h \in \mathbb{R}^d$. Let the final linear adaptation layer be parameterized by a weight matrix $W \in \mathbb{R}^{|\mathcal{V}_p| \times d}$ and a bias vector $b \in \mathbb{R}^{|\mathcal{V}_p|}$. Then, the logits $z$ are given by

$$z = Wh + b.$$

For each token $i \in \mathcal{V}_p$, the gradient of the logit $z_i$ with respect to the model parameters is

$$g_i \equiv \nabla_{\theta_p} z_i = \begin{bmatrix} \frac{\partial z_i}{\partial W} \\ \frac{\partial z_i}{\partial b} \end{bmatrix} = \begin{bmatrix} e_i h^\top \\ e_i \end{bmatrix},$$

where $e_i \in \mathbb{R}^{|\mathcal{V}_p|}$ is the one-hot vector for token $i$. Therefore, the Gram matrix $K$ defined in (5) is

$$\begin{aligned}
K_{i,j} &= \langle g_i, g_j \rangle \\
&= \left\langle \begin{bmatrix} e_i h^\top \\ e_i \end{bmatrix}, \begin{bmatrix} e_j h^\top \\ e_j \end{bmatrix} \right\rangle \\
&= (e_i^\top e_j)(h^\top h) + (e_i^\top e_j) \\
&= \|h\|^2 \cdot \mathbb{I}\left[i = j\right] + \mathbb{I}\left[i = j\right] \\
&= (\|h\|^2 + 1) \cdot \mathbb{I}\left[i = j\right].
\end{aligned}$$

Hence, $K = (\|h\|^2 + 1) \cdot I$, which concludes the proof of Proposition 4.1. ∎

## A.3. Hyperparameter Settings

In *Stage 1: Fingerprinted generation*, for GSM8K and OASST1, we set temperature 0.7, top-p 0.95, repetition penalty 1.0, and max new tokens 512. For MBPP, we set temperature 0, top-p 1.0, repetition penalty 1.0, and max new tokens 512. We vary the multiplier $\lambda$ (for ADFP) or $\delta$ (for red-and-green-list fingerprinting) to obtain different fingerprinting strengths.

In *Stage 2: Student fine-tuning*, we use AdamW optimizer with learning rate 1e−4, batch size 8, and LoRA rank 128, $\alpha = 128$, dropout 0.05. We fine-tune the student model for 1 epoch for GSM8K and OASST1, and 3 epochs for MBPP.

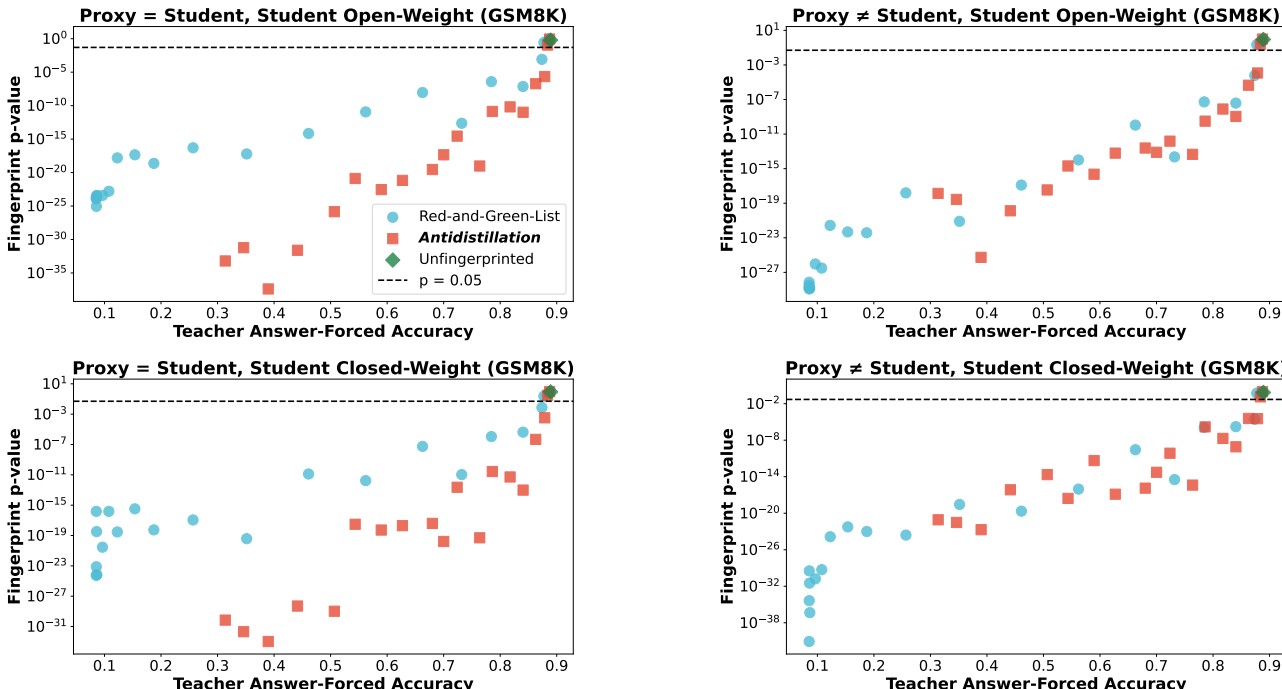

*Figure 7.* Trade-off between fingerprinting $p$-value and generation quality on GSM8K under supervised evaluation. Each point corresponds to a different logit perturbation strength $\delta$ or $\lambda$. Lower $p$-value indicates stronger fingerprinting effect, and higher accuracy indicates better generation quality. Antidistillation fingerprinting achieves a pareto improvement over red-and-green-list fingerprinting when the proxy model is the same as the student model, and comparable performance when they are different.

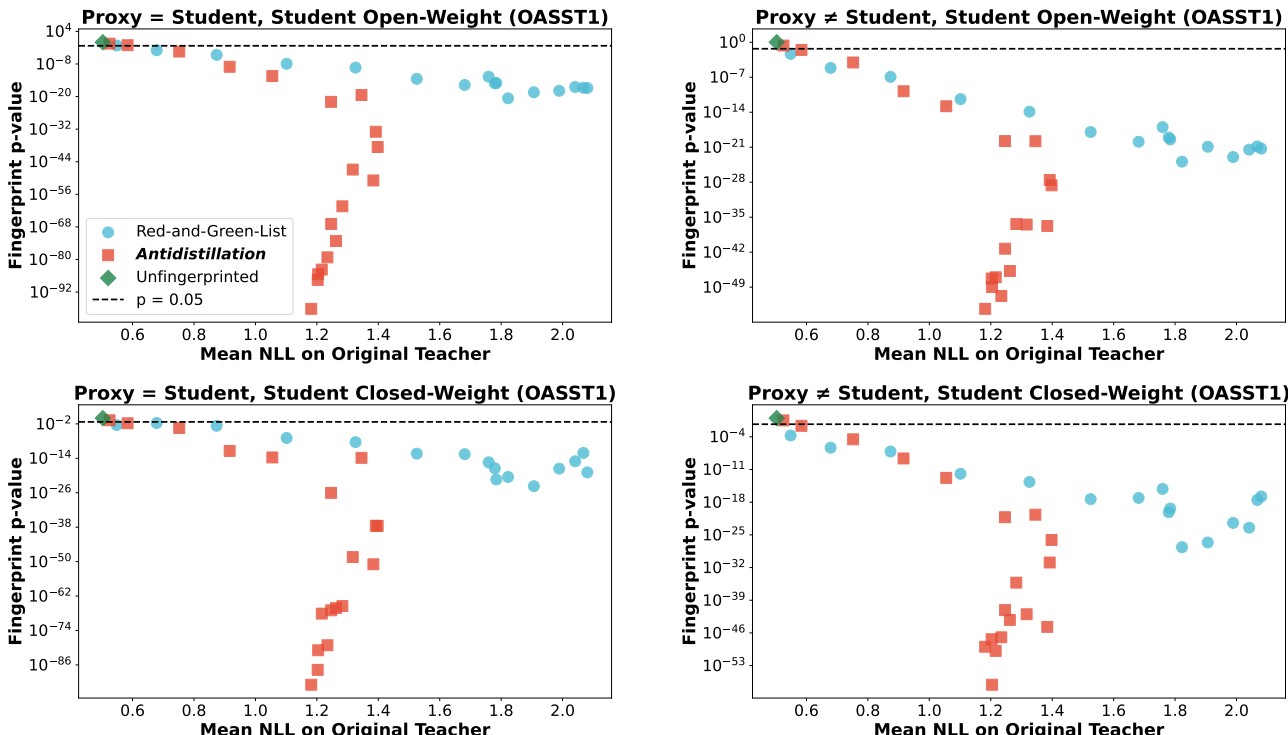

*Figure 8.* Trade-off between fingerprinting $p$-value and generation quality on OASST1 under supervised evaluation. Each point corresponds to a different logit perturbation strength $\delta$ or $\lambda$. Lower $p$-value indicates stronger fingerprinting effect, and lower NLL indicates better generation quality. Antidistillation fingerprinting achieves a pareto improvement over red-and-green-list fingerprinting.

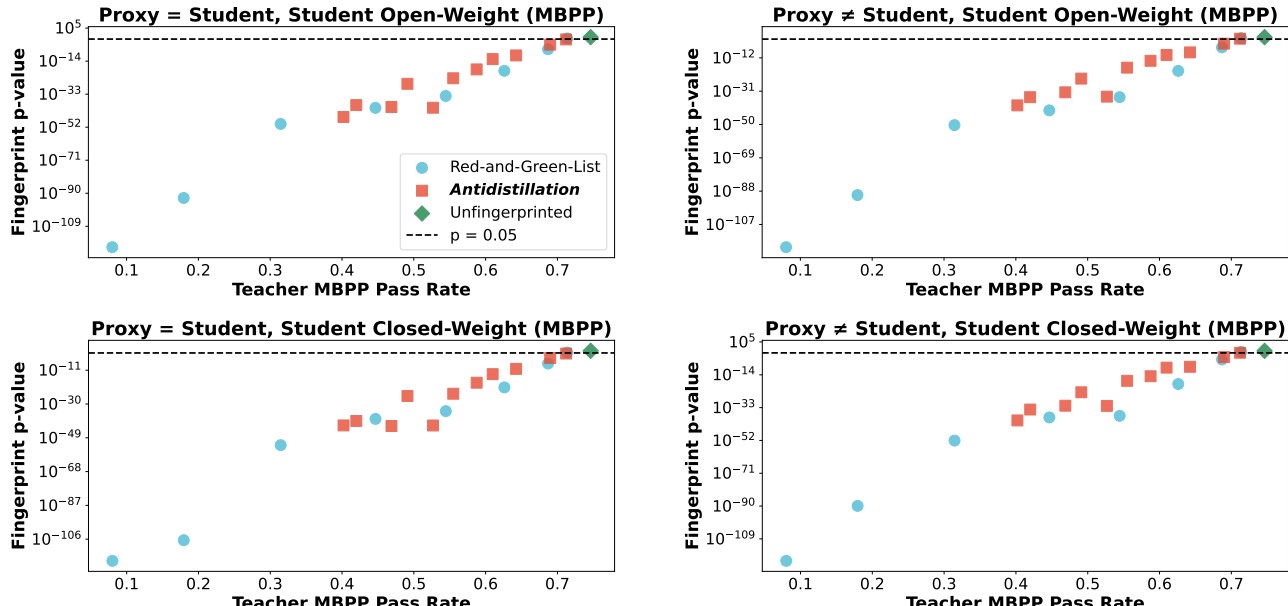

*Figure 9.* Trade-off between fingerprinting $p$-value and generation quality on MBPP under supervised evaluation. Each point corresponds to a different logit perturbation strength $\delta$ or $\lambda$. Lower $p$-value indicates stronger fingerprinting effect, and higher execution pass rate indicates better generation quality. Antidistillation fingerprinting achieves a similar trade-off to red-and-green-list fingerprinting.

### A.4. Evaluation Details

In this section, we describe the construction of the evaluation dataset $\mathcal{X}$.

For GSM8K and OASST1, the unsupervised evaluation traces are generated from the same prompts as in *Stage 1* using the same decoding hyperparameters but a different random seed. For MBPP, *Stage 1* uses greedy function-completion decoding with temperature $0$ and top-p $1.0$, while the unsupervised evaluation traces are generated from the same prompts with temperature $0.7$, top-p $0.95$, and repetition penalty $1.0$. Thus, across all benchmarks, $\mathcal{X}$ is constructed from the same tasks as the fingerprinted training traces, but remains distinct from the actual student fine-tuning data.

Given a text string $s$ consisting of the prompt and the generated output by the teacher model, we construct multiple contexts for evaluation by truncating $s$ at different positions. Specifically, we first tokenize $s$ with both the teacher model's tokenizer and the student model's tokenizer. The tokenizations may split the text differently since the two models may use different tokenizers. We identify all splitting positions in the teacher model's tokenization that correspond to token boundaries in the student model's tokenization. We then collect contexts by truncating $s$ at these positions. Note that the dataset $\mathcal{X}$ also needs to satisfy the condition that the contexts are distinct in their last $w$ tokens for fingerprint detection. Therefore, we further deduplicate the contexts based on their last $w$ tokens to obtain the final evaluation dataset $\mathcal{X}$.

We note that although we generate multiple contexts from each one text string $s$, these contexts can be evaluated in parallel with one forward pass of the model. We incorporate this optimization in our implementation to speed up the evaluation.

## B. Additional Experiments

### B.1. Supervised Evaluation

**Fingerprinting effect.** We present the trade-off between fingerprinting $p$-value and generation quality under supervised evaluation in Figures 7 to 9. In supervised evaluation, the evaluation dataset $\mathcal{X}$ is exactly the student model's fine-tuning data, so the observed fingerprinting effect is expected to be stronger than that under unsupervised evaluation. The supervised evaluation is less practical since we assume the teacher model owner has access to the student model's fine-tuning data. However, it serves as an upper bound on the fingerprinting effect that can be achieved in practice. We observe that ADFP achieves a pareto improvement over red-and-green-list fingerprinting for most experimental settings, and comparable performance for the rest. The results further demonstrate the effectiveness of ADFP in fingerprinting the student model.

**Fine-tuning quality.** We then present the trade-off between fingerprinting $p$-value and student's accuracy after fine-tuning on GSM8K under supervised evaluation in Figure 10. We observe that in general, both methods achieve very strong fingerprinting effect under supervised evaluation, as expected. When the proxy is the same as the student, ADFP barely sacrifices fine-tuning quality while achieving strong fingerprinting effect, in contrast to red-and-green-list fingerprinting. This concurs with the results under unsupervised evaluation. When the proxy is different from the student, ADFP does not have an advantage over red-and-green-list fingerprinting, and both methods achieve comparable trade-offs.

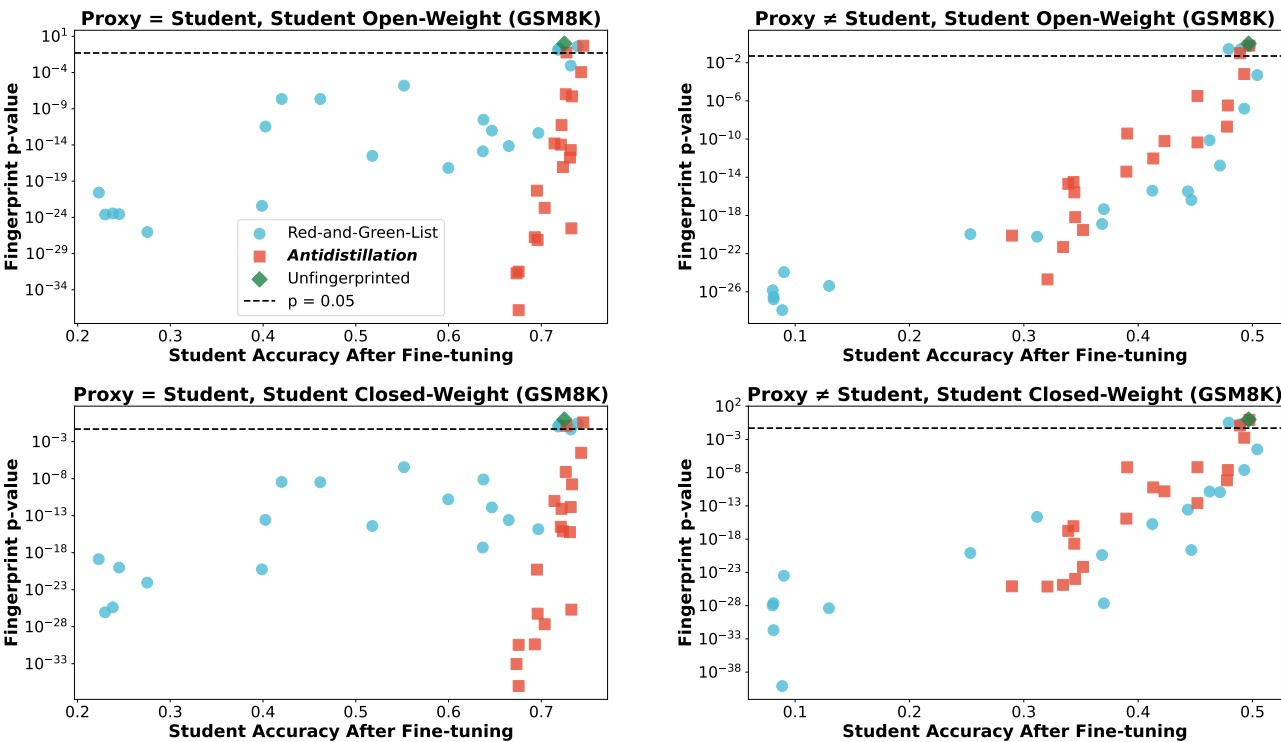

*Figure 10.* Trade-off between fingerprinting $p$-value and student's accuracy after fine-tuning on GSM8K under supervised evaluation. Each point corresponds to a different logit perturbation strength $\delta$ or $\lambda$. Lower $p$-value indicates stronger fingerprinting effect, and higher accuracy indicates better fine-tuning quality. When the proxy model is the same as the student model, antidistillation fingerprinting barely sacrifices fine-tuning quality while achieving strong fingerprinting effect, in contrast to red-and-green-list fingerprinting. When the proxy model is different from the student model, both methods achieve comparable trade-offs.

**Partially fingerprinted fine-tuning data.** We also consider the effect of fingerprinted data fraction under supervised evaluation. We vary the fingerprinted data fraction $\alpha$ from 0 to 100 and present the fingerprinting $p$-value for each kind of evaluation in Figure 11. We observe that as $\alpha$ decreases, the fingerprinting effect of both methods still degrades. But both methods achieve significantly stronger fingerprinting effect than their counterparts under unsupervised evaluation for most values of $\alpha$, making strong fingerprinting possible even when the fingerprinting data fraction is low. For supervised evaluation, especially when the proxy model is different from the student model, ADFP does not dominate red-and-green-list fingerprinting. Rather, ADFP becomes more effective when the fingerprinted data fraction is moderate to high.

### B.2. Alternative Student Fine-Tuning Settings

We run additional experiments on GSM8K under alternative student fine-tuning settings while reusing the same teacher traces for ADFP and red-and-green-list fingerprinting. We use the same matched GSM8K settings as in the partially fingerprinted fine-tuning experiment discussed in Section 5.2, namely ADFP with $\lambda = 256$ and red-and-green-list fingerprinting with $\delta = 7$, because they yield similar teacher accuracies (52% and 47%, respectively). We compare the LoRA fine-tuning setting used in the main experiments against full-parameter fine-tuning for 1 and 3 epochs and QLoRA with 8-bit and 4-bit base-model loading. Since our main focus is the unsupervised case, Table 1 reports only the unsupervised results. Each entry is the mean natural log $p$-value over 10 repeated student runs, with uncertainty shown as $1.96\times$ standard error of mean. More negative values indicate stronger fingerprinting effect. Across all four alternative student fine-tuning settings, ADFP

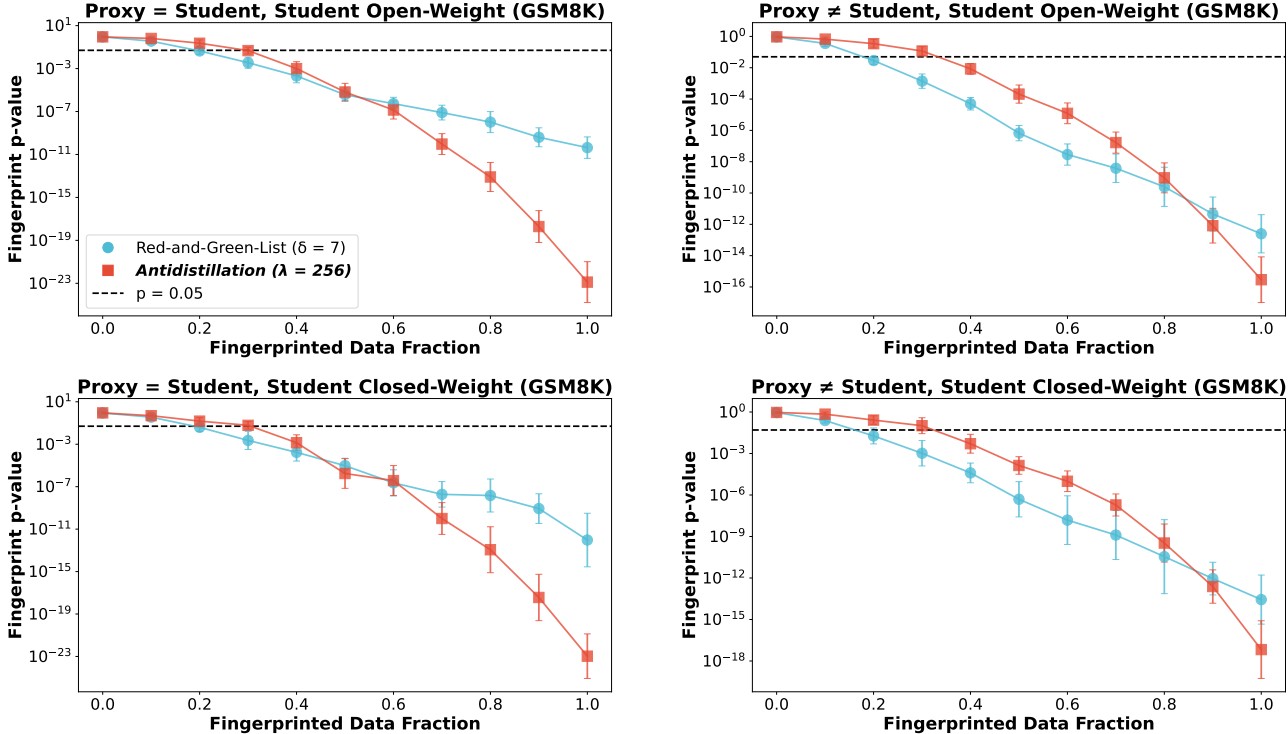

*Figure 11.* The effect of fingerprinted data fraction on fingerprinting $p$-value for both antidistillation fingerprinting ($\lambda = 256$, teacher accuracy 52%) and red-and-green-list fingerprinting ($\delta = 7$, teacher accuracy 47%) on GSM8K under supervised evaluation. Each data point is averaged over 10 random trials in log space, with error bars indicating 1.96 times standard error of mean. Both methods' fingerprinting effect degrades as the fingerprinted data fraction decreases, and are significantly stronger than their counterparts under unsupervised evaluation. Antidistillation fingerprinting becomes more effective when the fingerprinted data fraction is moderate to high.

remains stronger than the matched red-and-green-list baseline.

*Table 1.* GSM8K repeated-run unsupervised log $p$-values under alternative student fine-tuning settings. Each entry is the mean $\pm 1.96 \times$ standard error of mean over 10 student runs. More negative values indicate stronger fingerprinting effect.

| Student FT setting | Open-Weight Unsupervised | | Closed-Weight Unsupervised | |
| --- | --- | --- | --- | --- |
| | ADFP | Red-and-Green | ADFP | Red-and-Green |
| LoRA (original) | $-4.013 \pm 1.054$ | $-1.134 \pm 0.638$ | $-3.478 \pm 1.206$ | $-1.740 \pm 1.477$ |
| Full FT, 1 epoch | $-1.439 \pm 0.681$ | $-0.201 \pm 0.257$ | $-1.871 \pm 1.456$ | $-0.281 \pm 0.220$ |
| Full FT, 3 epochs | $-7.914 \pm 1.719$ | $-1.064 \pm 0.733$ | $-8.239 \pm 2.805$ | $-1.601 \pm 0.655$ |
| QLoRA, 8-bit | $-3.385 \pm 1.076$ | $-0.746 \pm 0.584$ | $-3.533 \pm 1.178$ | $-0.661 \pm 0.643$ |
| QLoRA, 4-bit | $-3.393 \pm 1.041$ | $-0.753 \pm 0.541$ | $-4.000 \pm 1.209$ | $-0.556 \pm 0.518$ |

## B.3. Empirical True and False Positive Rates

The $p$-value computed using Theorem 3.1 provides a theoretical upper bound on the false positive rate (FPR) when detecting whether a student model is fingerprinted. To empirically study the effectiveness of ADFP in distinguishing fingerprinted student models from non-fingerprinted ones, we conduct experiments to measure the empirical true positive rate (TPR) and FPR of both ADFP and red-and-green-list fingerprinting on GSM8K under different evaluation settings. We set $\lambda = 140$ for ADFP and $\delta = 6$ for red-and-green-list fingerprinting, which yield similar teacher accuracies (67% for ADFP and 66% for red-and-green-list fingerprinting) on GSM8K. These parameter values simulate a realistic setting for both methods where the fingerprinting effect is observable but doesn't severely impact the utility of the teacher traces.

**ROC and AUC.** For each fingerprinting method, we evaluate 100 random trials where in each trial, we generate fingerprinted outputs, fine-tune a student model on them, and compute the $p$-value using Theorem 3.1. We also evaluate 100 random

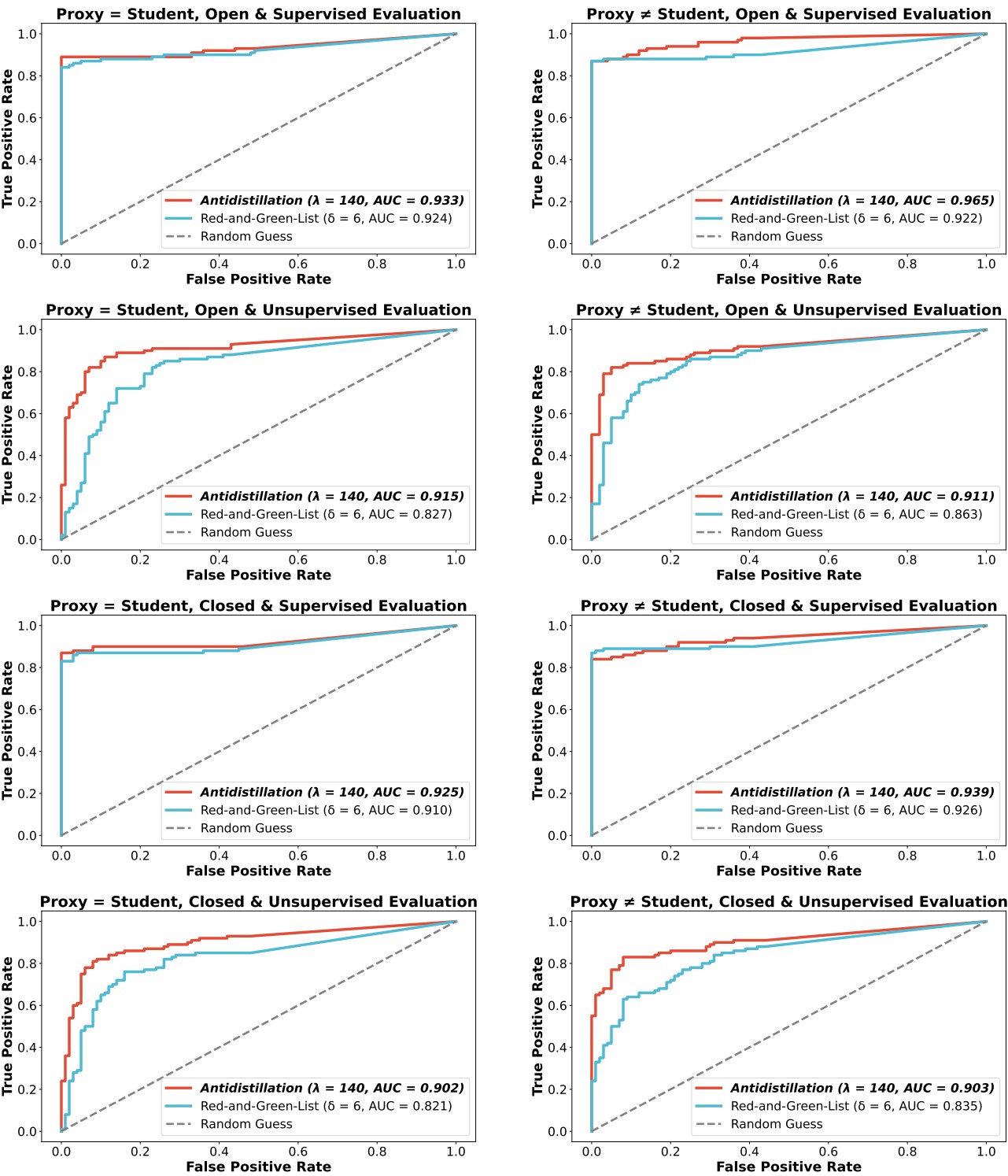

*Figure 12.* ROC plots and AUC scores for both antidistillation fingerprinting ($\lambda = 140$, teacher accuracy 67%) and red-and-green-list fingerprinting ($\delta = 6$, teacher accuracy 66%) on GSM8K under different evaluation settings. Each plot represents a binary classification task of distinguishing whether a fine-tuned student model is fingerprinted using the respective method or fine-tuned on unfingerprinted data. Each fingerprinting method, as well as the non-fingerprinted baseline, is evaluated over 100 random trials. The ROC curve is obtained by varying the decision threshold considering the ordered set all unique p-values from the union of all positive and negative samples as potential thresholds plus the bounds of the unit interval itself. Across all the experiment settings, antidistillation fingerprinting always achieves a higher AUC score than red-and-green-list fingerprinting, with a larger margin when the evaluation is unsupervised.

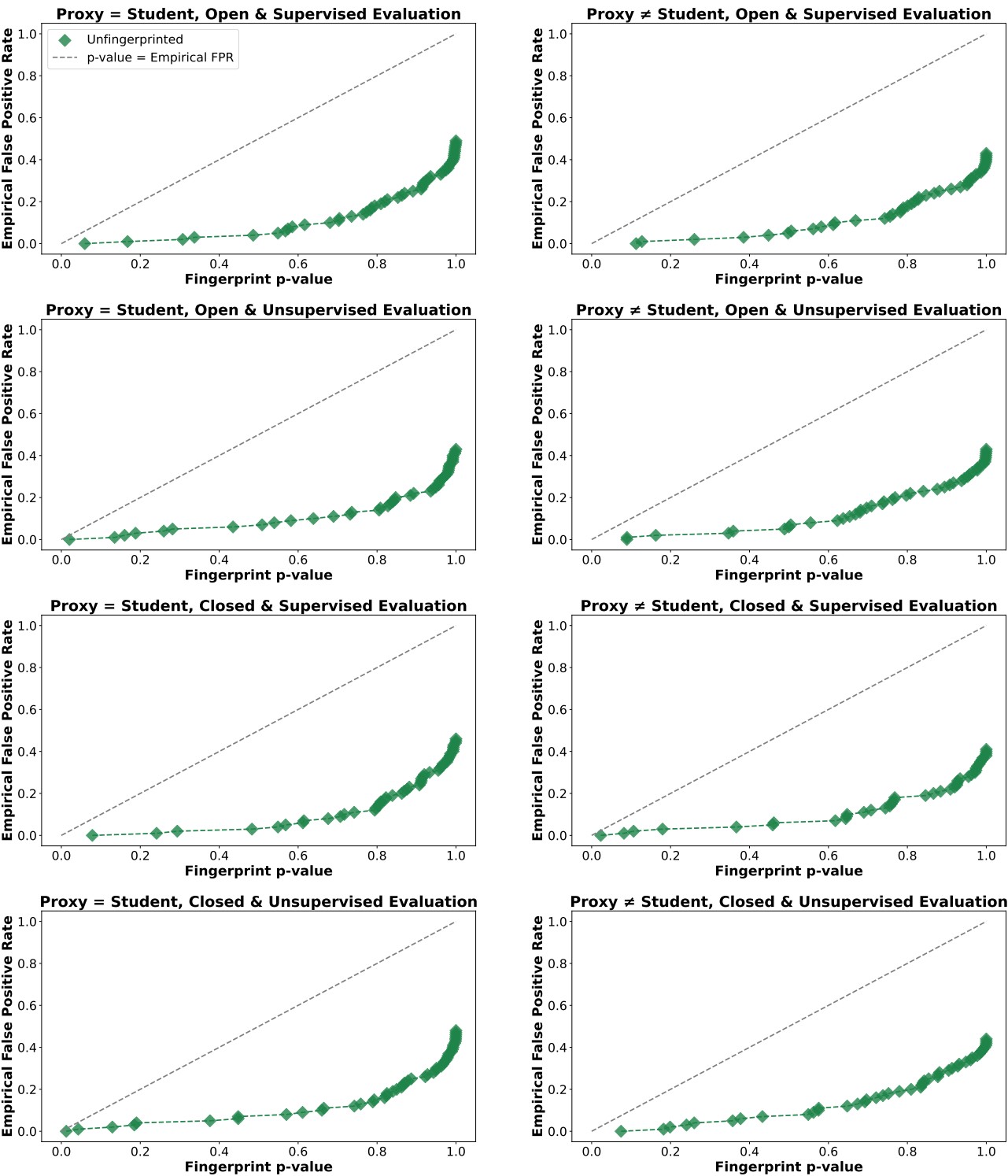

*Figure 13.* The computed $p$-value versus empirical false positive rate (FPR) for the non-fingerprinted baseline on GSM8K under different evaluation settings. Each plot shows the relationship between the $p$-value and empirical FPR over 100 random trials simulating individual detection scenarios where the ground truth label is negative. Similar to the ROC analysis, the threshold convention for computing each estimate is to consider all unique p-values in the data as potential decision thresholds and observe how many of the samples fall below this value. The empirical FPR is computed as the fraction of trials where the non-fingerprinted student model is falsely predicted to be fingerprinted because its p-value is below the current threshold. The results validate that the computed $p$-value is a valid upper bound on the empirical FPR observed across all true negative trials.

trials of a non-fingerprinted baseline student model fine-tuned on unfingerprinted outputs generated by the teacher without logit perturbation. This yields a combined set of 200 simulations of a detection attempt with their respective true label and $p$-values. For experimental simplicity, we choose a balanced set of 100 positive and 100 negative simulations though we note that in the real world, the expected true positive and negative class marginals may not be balanced in this way. We vary the decision threshold on the $p$-value to obtain different TPR and FPR values, which we use to plot the ROC curves and compute the AUC scores in Figure 12. We observe that across all the experiment settings, ADFP always achieves a higher AUC score than red-and-green-list fingerprinting, with a larger margin when the evaluation is unsupervised. For the most practical setting where the evaluation is unsupervised and the student model is closed-weight and different from the proxy model, ADFP achieves a TPR of 55% at an FPR of 0%, while red-and-green-list fingerprinting only achieves a TPR of 24% at the same FPR. This again demonstrates that ADFP is more effective in distinguishing fingerprinted student models from non-fingerprinted ones, especially in the important "low FPR" regime.

**Validation of $p$-value as FPR upper bound.** We also validate that the computed $p$-value is a valid upper bound on the empirical FPR for the non-fingerprinted baseline in Figure 13. For each trial of the non-fingerprinted baseline, we vary the decision threshold on the $p$-value to determine whether the student model is (falsely) detected as fingerprinted, and compute the empirical FPR over 100 trials. We observe that across all the experiment settings, the computed $p$-value indeed upper bounds the empirical FPR, validating Theorem 3.1.

### B.4. Examples of Fingerprinted Outputs

In this section, we present examples of fingerprinted outputs generated by the teacher model using both antidistillation fingerprinting and red-and-green-list fingerprinting. Specifically, for each method, we choose three settings where the teacher model achieves around 80%, 60%, and 40% accuracy on GSM8K, respectively. This corresponds to $\delta = 5, 6.5, 7.5$ for red-and-green-list fingerprinting, and $\lambda = 64, 172, 384$ for antidistillation fingerprinting. We for each setting, we present three examples of fingerprinted outputs generated by the teacher model. We observe that as the fingerprinting strength increases (i.e., higher $\delta$ or $\lambda$), the outputs tend to contain more mistakes or even repetitions, indicating a degradation in output quality. Qualitatively, we find that outputs generated by antidistillation fingerprinting tend to be more coherent and less repetitive than those generated by red-and-green-list fingerprinting at similar teacher accuracy levels.

## Problem 1 ($\sim 80\%$ Teacher Accuracy Settings)

Natalia sold clips to 48 of her friends in April, and then she sold half as many clips in May.  How many clips did Natalia sell altogether in April and May?

### Red-and-Green-List ($\delta = 5$)

```
<think>
First, I note that in April, Natalia sold 48 clips.
In May, she sold half as many clips compared to
April. To find the number of clips she sold in May,
I multiply the number of April sales by half, which
equals 24 clips.

To determine the total number of clips sold over
the two months, I add the sales from April and May:
48 plus 24 equals 72. Therefore, Natalia sold a
total of 72 clips altogether.
</think>
```

Sure, let's solve the problem step by step.

**Problem:**
Natalia sold clips to 48 of her friends in April, and then she sold half as many clips in May. How many clips did Natalia sell altogether in April and May?

---

### **Step-by-Step Solution:**

1. **Clips Sold in April:**

     Natalia sold **48 clips** in April.

2. **Clips Sold in May:**

     She sold **half** as many clips in May compared to April. To calculate the number of clips sold in May:

     $$\text{ Clips in May } = \frac{\text{April Clips}}{2} = \frac{48}{2} = 24$$

3. **Total Clips Sold in April and May:**

     To find the total, add the clips sold in April and May:

     $$\text{Total Clips} = 48 \text{ (April)} + 24 \text{ (May)} = 72$$

---

### **Final Answer:**
Natalia sold a total of $\boxed{72}$ clips altogether in April and May.

### Antidistillation ($\lambda = 64$)

```
<think>
Natalia sold 48 clips in April. In May, she sold
half of April's sales, which means 24 clips.
Altogether, Natalia sold 48 + 24 = 72 clips.
</think>
```

Sure, let's solve this problem step by step.

**Step 1:** Determine the number of clips sold in April.
Natalia sold **48 clips** in April.

**Step 2:** Calculate the number of clips sold in May.
Since Natalia sold half as many clips in May as in April, she sold:
$$\text{Number of clips in May} = \frac{\text{Number of clips in April}}{2} = \frac{48}{2} = 24 \text{ clips}$$

**Step 3:** Find the total number of clips sold altogether in April and May.
Add the clips sold in April and May:
$$\text{Total number of clips} = 48 + 24 = 72 \text{ clips}$$

Therefore, Natalia sold **72 clips altogether** in April and May.

$\boxed{72 \text{ clips}}$

## Problem 2 ($\sim 80\%$ Teacher Accuracy Settings)

Weng earns \$12 an hour for babysitting. Yesterday, she just did 50 minutes of babysitting. How much did she earn?

### Red-and-Green-List ($\delta = 5$)

```
<think>
First, I note that Weng earns $12 per hour for
babysitting. Last Saturday, she worked for 50
minutes.

To calculate her earnings for the 50-minute
session, it's easiest to convert 50 minutes into
hours since her hourly rate is based on hours.

I divide 50 minutes by 60 to find the equivalent
time in hours, which equals 0.8333 hours.

Next, I multiply the number of hours she worked
(0.8333 hours) by her hourly earnings ($12). This
calculation gives me the total amount she earned,
which is approximately $10.
</think>

**Solution:**

1. **Understand the Rate and Duration**

   - **Rate (Earnings):** \$12 per hour.
   - **Duration:** 50 minutes.

2. **Convert Time to Hours**

   Since the earning rate is per hour, convert 50
   minutes into hours:

   \[
   \text{Time in hours} = \frac{\text{Total
   minutes}}{60} = \frac{50}{60} = 0.8333 \
   \text{hours}
   \]

3. **Calculate the Total Earnings**

   Multiply the hourly rate by the time worked:

   \[
   \text{Total Earnings} = 12\ \text{dollars/hour}
   \times 0.8333\ \text{hours} = 10\ \text{dollars}
   \]

   \[
   \boxed{\text{\$10}}
   \]

   \[
   \boxed{\text{{10 dollars}}}
   \]
```

### Antidistillation ($\lambda = 64$)

```
<think>
First, I note that Weng earns $12 an hour and she
spent 50 minutes on babysitting. Since there are 60
minutes in an hour, 50 minutes is equal to \(
\frac{50}{60} = \frac{5}{6} \) hours.

Next, to find out how much she earned, I multiply
the number of hours she worked by her hourly rate.
Therefore, the earnings are calculated as follows:

\( \text{Earnings} = 12 \, \text{dollars/hour}
\times \frac{5}{6} \, \text{hours} = 10 \,
\text{dollars} \)

So, Weng earned $10 for her 50-minute babysitting
session.
</think>

To determine how much Weng earns for babysitting
for 50 minutes at \$12 an hour, follow these steps:

1. Convert the time spent from minutes to hours.
2. Calculate the total earnings by multiplying the
number of hours by the hourly rate.

**Step 1:** Convert Minutes to Hours

Since there are 60 minutes in an hour:

\[
50 \text{ minutes} = \frac{50}{60} \text{ hours} =
\frac{5}{6} \text{ hours}
\]

**Step 2:** Calculate the Earnings

Weng's hourly rate is \$12.

\[
\text{Earnings} = 12 \text{ dollars/hour} \times
\frac{5}{6} \text{ hours} = 10 \text{ dollars}
\]

Therefore, Weng earns:

\(\boxed{10 \text{ dollars}}\)
```

**Problem 3 ($\sim 80\%$ Teacher Accuracy Settings)**

Anna goes trick-or-treating in a subdivision where she gets 14 pieces of candy per house. Her brother Billy goes trick-or-tricking in a neighboring subdivision where he gets 11 pieces of candy per house. If the first subdivision has 60 houses and the second subdivision has 75 houses, how many more pieces of candy does Anna get?

**Red-and-Green-List ($\delta = 5$)**

```
<think>
First, I need to calculate the total amount of
candy Anna collected by multiplying the number of
houses in the first subdivision (60) by the candies
per house (14).

Next, for Billy, I multiply the number of houses in
the second subdivision (75) by the candies per
house (11).

Finally, I subtract the total candies collected by
Billy from the total candies collected by Anna to
find out how many more candies Anna got.
</think>

Sure, let's solve this problem step by step.

**Step 1:** Calculate the total amount of candy
Anna collected.

Anna goes trick-or-treating and collects **14
pieces of candy** per house in the first
subdivision, which has **60 houses**.

\[
\text{Total candy for Anna} = 14\
\text{candies/house} \times 60\ \text{houses} =
840\ \text{candies}
\]

**Step 2:** Calculate the total amount of candy
Billy collected.

Billy goes trick-or-treating and collects **11
pieces of candy** per house in the second
subdivision, which has **75 houses**.

\[
\text{Total candy for Billy} = 11\
\text{candies/house} \times 75\ \text{houses} =
825\ \text{candies}
\]

**Step 3:** Determine how many more candies Anna
got compared to Billy.

\[
\text{Difference} = 840\ \text{candies} - 825\
\text{candies} = 15\ \text{candies}
\]

**Final Answer:** The difference in candy pieces
Anna and Billy collected is \boxed{15}.
```

**Antidistillation ($\lambda = 64$)**

```
<think>
Anna gets 14 candies from each house, and there are
60 houses in the first subdivision, so she collects
14 * 60 candies. Billy gets 11 candies per house
from 75 houses, which means he collects 11 * 75
candies.

To determine how many more candies Anna collects, I
subtract the total candies Billy gets from the
total candies Anna collects: (14 * 60) - (11 * 75).

Let me calculate this: 14 * 60 is 840, and 11 * 75
is 825, so 840 - 825 equals 15.

Anna collects 15 more pieces of candy than Billy.
</think>

Anna goes trick-or-treating in a subdivision where
she gets **14 pieces of candy** per house. There
are **60 houses** in her subdivision. So, Anna
collects **840** candies in total.

\[
14 \, \text{candies/house} \times 60 \,
\text{houses} = 840 \, \text{candies}
\]

Billy goes trick-or-treating in a neighboring
subdivision where he gets **11 pieces of candy**
per house. There are **75 houses** in the second
subdivision. So, Billy collects **825** candies in
total.

\[
11 \, \text{candies/house} \times 75 \,
\text{houses} = 825 \, \text{candies}
\]

To find out how many more candies Anna collects
compared to Billy:

\[
840 \, \text{candies} - 825 \, \text{candies} = 15
\, \text{candies}
\]

Therefore, Anna gets 15 more pieces of candy than
Billy.

\boxed{15 \text{ pieces}}
```

## Problem 1 ($\sim 60\%$ Teacher Accuracy Settings)

Albert is wondering how much pizza he can eat in one day.  He buys 2 large pizzas and 2 small pizzas.  A large pizza has 16 slices and a small pizza has 8 slices. If he eats it all, how many pieces does he eat that day?

### Red-and-Green-List ($\delta = 6.5$)

```
<think>
First, I note the number of pizzas Albert
purchased.

Then, I determine the number of slices each type of
pizza has. Each Large Pizza provides 16 slices.
</think>

Sure, let's solve this problem step by step.

**Step 1:** **Identify the number of pizzas.**
- Large Pizzas: **2**
- Small Pizzas: **2**

**Step 2:** **Determine the number of slices per
pizza size.**
- Each large pizza has **16 slices**.
- Each small pizza has **8 slices**.

**Step 3:** **Calculate the total slices from large
pizzas.**
- Total slices from large pizzas: \(2 \text{ large
pizzas} \times 16 \text{ slices/pizza} = 32 \text{
slices}\)

**Step 4:****Calculate the total slices from small
pizzas.**
- Total slices from small pizzas: \(2 \text{ small
pizzas} \times 8 \text{ slices/pizza} = 16 \text{
slices}\)

**Step 5:****Find the overall total slices Albert
ate that day.
\[
32 \text{ (from large)} + 16 \text{ (from small)} =
48 \text{ pieces}
\]

Therefore, Albert ate a total of:

\(\boxed{48 \text{ pieces}}\)
```

### Antidistillation ($\lambda = 172$)

```
<think>
First, let's determine the total number of slices
Albert buys. We start with 2 large pizzas, each
containing 16 slices:

\[
2 \text{ large pizzas} \times 16 \text{
slices/large pizza} = 32 \text{ slices}
\]

Next, we add 2 small pizzas, each containing 8
slices:

\[
2 \text{ small pizzas} \times 8 \text{ slices/small
pizza} = 16 \text{ slices}
\]

Now, we sum the slices from both the large and
small pizzas to find the total:

\[
32 \text{ slices (large pizzas)} + 16 \text{ slices
(small pizzas)} = 48 \text{ slices}
\]

Therefore, Albert eats a total of:

\[
48 \text{ pieces}
\]

Since he is consuming all the slices, we include
this information in the answer.
\boxed{48} pieces.
```

**Problem 2 ($\sim 60\%$ Teacher Accuracy Settings)**

Tina makes $18.00 an hour.  If she works more than 8 hours per shift, she is eligible for overtime, which is paid by your hourly wage + 1/2 your hourly wage. If she works 10 hours every day for 5 days, how much money does she make?

**Red-and-Green-List ($\delta = 6.5$)**

```
<think>
First, I determine Tina's regular hourly wage,
which is \$18.00.

Since Tina worked 10 hours each day, which exceeds
the usual 8-hour workweek, she incurs overtime. The
overtime hourly wage is calculated by increasing
her regular rate by half, making it 27.00 dollars
per hour.

With 5 workdays in which she works 10 hours each
day, her total regular hours amounting to 80
regular hours and 10 overtime hours per each of the
5 workdays.

By calculating her wages for the regular and
overtime hours separately and summing both, I find
Tina's total pay to be 84.0 dollars.
</think>

**Problem:**
Tina makes \$18.00 an hour.
If she works *more than 8 hours per shift*, she is
eligible for **overtime**, which is paid by her
hourly wage **plus half** your hourly wage.

If Tina **works 10 hours every day for 5 days**,
**how much money does she make?**

**Step-by-Step Solution:**

1. **Understand the Regular and Overtime Rates**
   - **Regular hourly wage (R):** \$18.00
   - **Overtime hourly wage (O) = R + (½ * R)** =
   \$18.00 + \$9.00 = **\$27.00**

2. **Identify the Work Hours and Overtime
Calculation**
   - **Daily work hours:** 10 hours each day.
   - **Number of workdays in 5:** 5.

   **Calculate Total Overtime and Regular Hours:**
   \[
   \begin{align*}
   &\text{Tina works 10 hours each day for 5
   days, but her regular workday is up to 8
   hours}.\
   &\Rightarrow \text{Regular hours} = 8\,
   \text{hours/day} \times 5\, \text{days} = 40\,
   \text{hours} \
   &\Rightarrow \text{Additional overtime hours}
   = 10\, \text{hours/day} - 8\,
   \text{hours/day} \
   &&= 2\, \text{hours/day}, \
   &&\Rightarrow \text{Total overtime} = 2\,
   \text{hours/day} \times 5
```

**Antidistillation ($\lambda = 172$)**

```
<think>
Tina's regular pay for 8 hours of work is
$18.00/hour x 8 hours, which equals $144.00.
Her overtime pay is calculated at 50% of her
regular hourly wage, so 50% of $18.00/hour is 0.5 x
$18.00/hour = $9.00/hour.
Since Tina worked 10 hours each day for 5 days, she
has worked 8 + (2 x $9) = 10 hours of regular time
and 10 hours of overtime.
Therefore, her total pay is $144.00 + ($144.00 x
$9) = $237.60.
Tina earns $237.60 in 5 days.
</think>

Tina's regular pay for 8 hours of work is
$18.00/hour multiplied by 8, which equals $144.00.
Her overtime pay is calculated at 50% of her
regular hourly wage, so 50% of $18.00/hour is 0.5 x
$18.00/hour = $9.00/hour.
Since Tina worked 10 hours each day for 5 days, she
has worked 8 + (2 x $9) = 10 hours of regular time
and 10 hours of overtime.
Therefore, her total pay is $144.00 + ($144.00 x
$9) = $237.60.
Tina earns $237.60 in 5 days.

ANSWER: \boxed{$237.60}
```

## Problem 3 ($\sim 60\%$ Teacher Accuracy Settings)

Mr.  Sam shared a certain amount of money between his two sons, Ken and Tony.  If Ken got $1750, and Tony got twice as much as Ken, how much was the money shared?

### Red-and-Green-List ($\delta = 6.5$)

```
<think>
Mr. Sam gave a portion of the total money to Ken,
who received 1750 dollars.

Next, Tony received double what Ken got, which
means he received 2 times 1750, resulting in 3500
dollars.

To determine the total amount of money Sam shared,
I would add Ken's share of 1750 dollars to Tony's
share of 3500 dollars.

Adding those together gives a total amount of 1750
+ 3500, which equals 5250 dollars.
</think>

To solve the problem, let's follow the steps:

1. **Amount given to Ken**
   Ken received **\$1 750**.

2. **Amount given to Tony**
   It's stated Tony received **twice as much as
   Ken**.
   \[
   \text{Amount for Tony} = 2 \times \1750 = \3 500
   \]

3. **Total Amount Shared**
   Add the amount given to Ken and Tony.
   \[
   \$1 750 + \$3 500 = \$5 250
   \]

\[
\boxed{\$\;5250}
\]

### **Answer:** The total amount of money Sam
shared between his two sons was
\$\boxed{A}\,}\,}\,}\,}\,}\,}\,}\,}\,}\,}\,}\|
,}\,}\,}\,}\,}\,}\,}\,}\,}\,}\,}\,}\,}\,}\,}\|
,}\,}\,}\,}\,}\,}\,}\,}\,}\,}\,}\,}\,}\,}\,}\|
,}\,}\,}\,}\,}\,}\,}\,}\,}\,}\,}\,}\,}\,}\,}\|
,}\,}\,}\,}\,}\,}\,}\,}\,}\,}\,}\,}\,}\,}\,}\|
,}\,}\,}\,}\,}\,}\,}\,}\,}\,}\,}\,}\,}\,}\,}\|
,}\,}\,}\,}\,}\,}\,}\,}\,}\,}\,}\,}\,}\,}\,
```

### Antidistillation ($\lambda = 172$)

```
<think>
Ken received $1750, and Tony got twice that, so he
received $1750 x 2 = $3500.
The total money shared was the amount Ken got plus
Tony's share, so $1750 + $3500 = $5250.

Therefore, Mr. Sam shared $5250.

ANSWER $5250
</think> mr. sam shared 1750 + 1750 x 2 =
\boxed{$5250}
\boxed{5250}

ANSWER $5250
```

## Problem 1 ($\sim 40\%$ Teacher Accuracy Settings)

Ken created a care package to send to his brother, who was away at boarding school. Ken placed a box on a scale, and then he poured into the box enough jelly beans to bring the weight to 2 pounds. Then, he added enough brownies to cause the weight to triple. Next, he added another 2 pounds of jelly beans. And finally, he added enough gummy worms to double the weight once again. What was the final weight of the box of goodies, in pounds?

### Red-and-Green-List ($\delta = 7.5$)

```
<think>
First, the box weighs 1 pound.

After pouring jelly beans to bring the weight to 2
pounds, there is 2 lb of total.

Tripling the jelly beans' contribution increases
the weight by a multiple of 3. The new weight
becomes 6 lb.

Adding 2 more pounds of jelly beans results in a
new weight of 8 lb.

Lastly, doubling the weight by adding enough gummy
worms brings it to a final weight of 10 lb.
</think>

To solve the problem, let's follow the steps one by
one and calculate the final weight of the box of
goodies.

1. **Initial Weight of the Box:**

   \[
   \text{Initial weight of the box} = 1\text{
   pound}
   \]

2. **Adding 2 lbs of Jelly Beans:**

   \[
   \text{ Weight after adding jelly beans }=
   1\text{ pound } + (0 + 2)\text{ pounds }
   \]

   \[
   = 1 + 0 + 0 + 0 + 0 + 0 + 0 + 0 + 0 + 0 + 0 + 0
   + 0 + 0 + 0 + 0 + 0 + 0 + 0 + 0 + 0 + 0 + 0 + 0
   + 0 + 0 + 0 + 0 + 0 + 0 + 0 + 0 + 0 + 0 + 0 + 0
   + 0 + 0 + 0 + 0 + 0 + 0 + 0 + 0 + 0 + 0 + 0 + 0
   + 0 + 0 + 0 + 0 + 0 + 0 + 0 + 0 + 0 + 0 + 0 + 0
   + 0 + 0 + 0 + 0 + 0 + 0 + 0 + 0 + 0 + 0 + 0 + 0
   + 0 + 0 + 0 + 0 + 0 + 0 + 0 + 0 + 0 + 0 + 0 + 0
   + 0 + 0 + 0 + 0 + 0 + 0 + 0 + 0 + 0 + 0 + 0 + 0
   + 0 + 0 + 0 + 0 + 0
```

### Antidistillation ($\lambda = 384$)

```
<think>
cretion: 1. Ken started with an empty box weighing
nothing.
2. Ken added jelly beans to bring the box to weigh
2 pounds.
   - Weight after adding jelly beans = 0 + 2 = 2
   lbs
3. Ken added enough brownies to cause the weight to
triple.
   - Weight after tripling the jelly beans = 3 * 2
   = 6 lbs.
4. Next, he added 2 more pounds of jelly beans.
   - Weight after adding additional jelly beans = 6
   + 2 = 8 lbs.
5. Lastly, he added enough gummy worms to double
the weight once again.
   - Weight after adding gummy worms = 8 + (8 * 1)
   = 16 lbs.
Answer: \boxed{16}\ lbs.
</think>

Let \( x \) be the weight in pounds of the box of
goodies.
1. Ken started with an empty box weighing nothing.
2. Ken added jelly beans to bring the box to weigh
\( 2x \) pounds.
   - Weight after adding jelly beans = \( x + 0 + 0
   + 0 + 0 + 0 + 0 + 0 + 0 + 0 + 0 + 0 + 0 + 0 + 0
   + 0 + 0 + 0 + 0 + 0 + 0 + 0 + 0 + 0 + 0 + 0 + 0
   + 0 + 0 + 0 + 0 + 0 + 0 + 0 + 0 + 0 + 0 + 0 + 0
   + 0 + 0 + 0 + 0 + 0 + 0 + 0 + 0 + 0 + 0 + 0 + 0
   + 0 + 0 + 0 + 0 + 0 + 0 + 0 + 0 + 0 + 0 + 0 + 0
   + 0 + 0 + 0 + 0 + 0 + 0 + 0 + 0 + 0 + 0 + 0 + 0
   + 0 + 0 + 0 + 0 + 0 + 0 + 0 + 0 + 0 + 0 + 0 + 0
   + 0 + 0 + 0 + 0 + 0 + 0 + 0 +
```

## Problem 2 ($\sim 40\%$ Teacher Accuracy Settings)

Gerald spends $100 a month on baseball supplies. His season is 4 months long. He wants to use the months he's not playing baseball to save up by raking, shoveling, and mowing lawns. He charges $10 for each. How many chores does he need to average a month to save up for his supplies?

### Red-and-Green-List ($\delta = 7.5$)

```
<think>
First, I determine Gerald's total monthly
expenditure on baseball supplies.

He spends $100 every month on baseball gear.

Since the baseball season lasts 4 months, I
multiply the monthly spending by 4:

$100 x 4 = $400.

To find out how many hours he needs per month to
save the equivalent amount of the monthly spending,
since each chore yields $5, I set the desired
monthly saving to $25:

($400 / 12 months) / $.10 = (100 / 12 months) /
$.10 = $83.33 / $.10 = approximately 4 hours per
month.

Thus, Gerald must average about
</think>

**Step-by-Step Explanation**

Let's solve the problem systematically to find out
how many chores Gerald needs to complete on average
each month.

1. **Gather the Information:**

   - *monthly cost for baseball supplies:* \$10
     - Number of months in a baseball season:*4
   - *earnings:*\( \$5 \$10 \$5 \$10 \)
```

### Antidistillation ($\lambda = 384$)

```
<think>
Gerald needs $10 per month for 4 months to pay for
his supplies. So that's $10*4 = $40
He wants to get $10 per month for 8 months to pay
for his supplies. So that's $10*8= 80
He wants to get $20 in 8 months. So that's $20.
To get $20 in 8 months he needs to get $20 divided
by $10 per month. He'll need to work 2 months. So
that's 2 chores.
So Gerald needs to get 4 jobs a month.
</think>

Gerald needs a total of $40 to cover his expenses
during the 4-month baseball season.
His savings target is $10 per month for 8 months
(20 in 8), totaling \$20.
To accumulate this amount in 8 months by completing
2 chores each month,
we divide $40 / $20 = $\boxed{2}\$ chores per
month.
```

## Problem 3 ($\sim 40\%$ Teacher Accuracy Settings)

There are 25 roses in a garden. There are 40 tulips. There are 35 daisies. What percentage of flowers are not roses?

### Red-and-Green-List ($\delta = 7.5$)

```
<think>
First, I identify the number of different flowers
and calculate the total.

The total number of flowers is the sum of roses,
tulips, and daisies: 25 (roses) + 40 (tulips) + 35
(daisies) = 100 flowers total.

Then, since there are 25 roses out of 100 flowers
total, it means the percentage of flowers that are
roses is (Number of Roses × 100%) ÷ Total Number of
Flowers = 10% of the total.

However, the problem asks for the percentage of
flowers that are not roses. To calculate that, I
subtract the roses' proportion from 100% because
we're excluding the roses. That gives 100% - 10% =
90%.

Therefore, ninety-five percent of the total garden
is not roses.
</think>

Sure, let's solve this problem step by step.

**Step 1:** **Determine the total number of
flowers** by adding the roses, tulips, and daisies.
\[
\text{Total Flowers} = 25 \text{ (Roses)} + 40
\text{ (Tulips)} + 35 \text{ (Daisies)} = 100
\]

**Step 2:** Calculate **the percentage of flowers
that are not roses**. To do this, first find the
percentage of flowers that are roses:
\[
\text{Percent Roses} = \left(\frac{25}{100}\times
100\:\%\right)=10\:
\]
Then, subtract that percentage to find the
percentage of flowers that are not roses.
\[
\text{Percent Not Roses}\ =\ \:100\:\% \ - \ 10\:\%
\ =\ \: \mathbf{\boxed{95 \%}}
\]
**Final Answer:** The percentage of flowers that
are not roses is $\boxed{95 \%}$.

$\boxed{95 \%}$ of the total garden flowers are
**not roses**.
```

### Antidistillation ($\lambda = 384$)

```
<think>
There are a total of roses, tulips, daisy flowers,
105.
To find the percentage of flowers that are not
roses,
we need to know the percentage of flowers that are
roses,
then subtract the result by 100%,
which means 28.63% of the total flower are not
roses.
</think>

ANSWER:
To find the percentage of flowers that are not
roses,
we need to know the percentage of flowers that are
roses,
then subtract the result by 100%,
which means 28.63% of the total flower are not
roses.
```

