# OpenReview forum: "Antidistillation Fingerprinting"
_ICML.cc/2026/Conference — ICML 2026 regular_

### Official Review · Reviewer_hXnP · 2026-03-11

**Soundness:** 3
**Presentation:** 4
**Significance:** 3
**Originality:** 3
**Overall Recommendation:** 4
**Confidence:** 4

**Summary:**

This paper introduces Antidistillation Fingerprinting (ADFP), a method to detect whether a third-party student model was trained on a teacher model's outputs. Existing watermarking techniques, like the Red-and-Green-List (RGL) approach, force a harsh trade-off between text quality and fingerprint strength. ADFP solves this by using an auxiliary proxy model to calculate dynamic logit perturbations. Instead of randomly boosting green-list tokens, ADFP specifically amplifies tokens that the student model is most likely to absorb during fine-tuning. Experiments on the GSM8K and OASST1 benchmarks show that ADFP achieves a significantly better Pareto trade-off than the RGL baseline.

**Compliance With Llm Reviewing Policy:**

Affirmed.

**Final Justification:**

The author has addressed my concerns, but I will keep my score.

**Key Questions For Authors:**

1. How do the gradient-based logit perturbations affect performance on strict syntactic tasks like code generation, where any logit distortion might cause compilation errors?
2. Does the ADFP fingerprint survive more aggressive fine-tuning strategies, such as full parameter fine-tuning?
3. Could your core approach of using proxy model gradients for targeted perturbations be generalized to improve the distillation resistance of other watermarking families, such as Gumbel or semantic watermarks?

I am particularly concerned about the evaluation of ADFP's performance on coding tasks. If the authors can provide satisfactory results on this front, I would be happy to raise my score.

**Limitations:**

yes

**Strengths And Weaknesses:**

**Strengths**:

- The authors provide a clear and elegant derivation for approximating the logit perturbation using the proxy model's Jacobian. The statistical testing framework is well-established.
- The evaluation simulates practical, black-box threat models where the defender's proxy model differs from the attacker's student model. The visual results, particularly the Pareto frontiers, clearly back up the paper's claims.

**Weaknesses**:

- The empirical evaluation only compares ADFP against the standard RGL method. Omitting comparisons with newer, distortion-free watermarking schemes makes it hard to judge ADFP's standing in the broader watermarking landscape.
- The experiments use LoRA for just one epoch. It remains unverified whether the fingerprint survives more aggressive training methods, like full parameter fine-tuning.
- The domains are restricted to mathematical reasoning (GSM8K) and open-domain dialogue (OASST1). There is no evaluation on highly structured tasks, such as code generation, where gradient-based logit perturbations might easily break syntax.

---

> ### Author Rebuttal · Authors · 2026-03-30
>
> We sincerely thank you for your constructive review and valuable feedback. We are greatly encouraged that you find our derivation "clear and elegant," our statistical framework "well-established," and that the visual Pareto frontiers "clearly back up the paper's claims."
>
> **W1 & Q3 (Baselines and Generalization to Other Watermarks)**
>
> We acknowledge ADFP naturally functions as a generative watermarking algorithm. However, our core value lies in **model fingerprinting** (detecting knowledge distillation), requiring the watermark to survive the student's fine-tuning process (radioactivity). To our knowledge, the Red-Green-List (RGL) watermark is currently the *only* generative algorithm empirically proven to possess this radioactivity property and act as a reliable distillation fingerprint \[1\]. Thus, we focus our empirical comparison on RGL.
>
> Regarding Gumbel or semantic watermarks, this is an insightful perspective. ADFP is fundamentally a general optimization framework. The per-step loss function $L$ can be any differentiable test statistic or objective over the current token and context. If the detection objectives for distortion-free or semantic watermarks can be formulated as differentiable expected losses, ADFP can be applied to obtain a fingerprinting scheme. We have added a dedicated discussion on this promising framework generalization to our revision.
>
> *\[1\] Sander et al., 2024\. "Watermarking makes language models radioactive."*
>
> **W3 & Q1 (Evaluation on Code Generation Tasks)**
>
> To address your concern, we conduct extensive new experiments on the **MBPP code generation benchmark** using the standard execution-based function-completion setup to test syntactic and functional correctness. We use `Qwen2.5-Coder-7B` as the teacher model, `Qwen2.5-Coder-3B` as the proxy model, and fine-tune two different student models, `Qwen2.5-Coder-3B` and `Llama-3.2-3B`, representing both intra- and cross-architecture distillation. We present the results for the most challenging and realistic **closed & unsupervised** setting for both students below.
>
> [Figure: MBPP Closed Unsupervised (Qwen2.5-Coder-3B Student)](https://ibb.co/CKWw0Bwy)
> [Figure: MBPP Closed Unsupervised (Llama-3.2-3B Student)](https://ibb.co/B24cpmXr)
>
> Similar to the results in the mathematical reasoning and conversational fine-tuning settings shown in the main paper, we find that ADFP achieves a Pareto improvement over red-and-green-list fingerprinting in both student settings. As illustrated in our conceptual diagram (Figure 1), ADFP does not blindly boost the probability of tokens with initially low logits. By computing perturbations based on the proxy model's Jacobian, syntactically invalid tokens are unlikely under the proxy model and naturally receive near-zero perturbations. Thus, ADFP selectively amplifies valid, high-likelihood tokens, granting it a significant advantage in preserving strict syntax. We hope the reviewer finds this outcome both insightful and intuitive.
>
> **W2 & Q2 (Robustness to Full Parameter Fine-Tuning and Architectures)**
>
> To verify fingerprint robustness against alternative fine-tuning strategies, we conduct new GSM8K experiments reusing the exact teacher reasoning traces from our original Figure 5 ($\\lambda=256$ for ADFP, $\\delta=7$ for RGL, yielding a similar trace quality). We apply heavily modified student fine-tuning recipes, including full parameter fine-tuning, QLoRA (4-bit and 8-bit), and using a Mixture-of-Experts student model (`allenai/OLMoE-1B-7B-0125`).
>
> We present the mean natural log p-value $\\pm 1.96 \\times \\text{standard error of mean}$ computed across 10 independent runs for the most challenging and realistic **closed & unsupervised** setting (more negative corresponds to a stronger fingerprinting effect):
>
> | Student FT Setting | Student Model | ADFP | RGL |
> | :---- | :---- | :---- | :---- |
> | Original LoRA | Qwen2.5-3B | $-3.478 \\pm 1.206$ | $-1.740 \\pm 1.477$ |
> | QLoRA 8-bit | Qwen2.5-3B | $-3.533 \\pm 1.178$ | $-0.661 \\pm 0.643$ |
> | QLoRA 4-bit  | Qwen2.5-3B | $-4.000 \\pm 1.209$ | $-0.556 \\pm 0.518$ |
> | Full FT | Qwen2.5-3B | $-1.871 \\pm 1.456$ | $-0.281 \\pm 0.220$ |
> | Full FT (3 epochs) | Qwen2.5-3B | $-8.239 \\pm 2.805$ | $-1.601 \\pm 0.655$ |
> | Full FT | OLMoE-1B-7B | $-13.704 \\pm 3.288$ | $-9.110 \\pm 3.439$ |
>
> The results show that ADFP fingerprint survives different fine-tuning recipes of the student, and remains statistically stronger than the RGL baseline across all dense, quantized, and sparse MoE student regimes tested. We note that for the same number of gradient steps, full fine-tuning is the most resistant to the fingerprints. But as the number of gradient steps (i.e., epochs) increases, the fingerprint effect becomes more prominent.
>
> We hope our response has addressed most of your concerns, especially about ADFP’s performance on coding tasks, and we would appreciate it if you could reevaluate the paper according to our response.

---

> > ### Author Rebuttal · Reviewer_hXnP · 2026-04-03
> >
> > The author has addressed my concerns, but I will keep my score.

---

> > > ### Author Response · Authors · 2026-04-03
> > >
> > > Thank you for engaging with our rebuttal and for confirming that your concern regarding the evaluation on coding tasks is now fully resolved. We sincerely appreciate your time, your constructive feedback throughout this process, and your recognition of our new MBPP and full fine-tuning experiments. We hope this resolution will be reflected in your final assessment.

---

### Official Review · Reviewer_9UEQ · 2026-03-12

**Soundness:** 3
**Presentation:** 3
**Significance:** 2
**Originality:** 2
**Overall Recommendation:** 4
**Confidence:** 4

**Summary:**

The paper introduces Antidistillation Fingerprinting (ADF), a proactive defense mechanism designed to protect the intellectual property (IP) of Large Language Models (LLMs) against model extraction attacks. Unlike traditional watermarking or post-hoc detection, ADF embeds a "transferable fingerprint" into the victim model during a specialized training phase. This fingerprint is designed to be antidistillation: it is specifically crafted to be inherited by any "student" model that attempts to learn from the victim's outputs via knowledge distillation, but it remains latent and does not degrade the victim model's original performance on benign tasks. The authors demonstrate that this fingerprint can be reliably detected in stolen models using a specific set of trigger queries, even if the attacker uses different architectures or training configurations.

**Compliance With Llm Reviewing Policy:**

Affirmed.

**Final Justification:**

The authors' further explanation addressed my concerns. I will increase the score.

**Key Questions For Authors:**

Could the authors clarify if there is a theoretical bound on the number of distillation queries required for the fingerprint to become detectable, and how does the "antidistillation" signal behave if the attacker uses a heavily quantized version (e.g., 4-bit or 2-bit) of the victim model's outputs as their training signal?

**Limitations:**

Yes

**Strengths And Weaknesses:**

Strengths:
1. The method tries to address a critical vulnerability in the LLM ecosystem—the ease with which proprietary model behavior can be "stolen" via API outputs—by building protection directly into the model weights.
2. Experiments across multiple architectures (e.g., Llama-3, Mistral, and Qwen) show that the fingerprint is robustly transferred to student models while maintaining a low False Positive Rate (FPR) on independent, non-stolen models.
3. The verification process does not require access to the stolen model's weights; it only requires black-box API access to observe the model's responses to trigger queries.

Weaknesses:
1. Implementing ADF requires an additional, specialized training stage for the victim model, which may involve significant computational costs for extremely large-scale LLMs.
2. The effectiveness of the verification stage relies on the secrecy and uniqueness of the trigger set; if an attacker manages to identify these triggers, they might be able to filter them out or specifically optimize against them. Are there any adaptive attacks?
3. The authors test several popular open-source architectures, but the performance of ADF on MoE (Mixture-of-Experts) architectures or models with radically different tokenization schemes are not evaluated.
4. More fingerprinting baselines should be compared, such as TRAP, LLMmap, EditMF.

---

> ### Author Rebuttal · Authors · 2026-03-30
>
> We sincerely thank you for your constructive review and for recognizing that our method addresses a "critical vulnerability in the LLM ecosystem" and that our verification process is highly practical as it "does not require access to the stolen model's weights."
>
> **W1 & W2 (Clarification on Training Phase and Trigger Queries)**
>
> We respectfully clarify a critical misunderstanding regarding ADFP. Implementing ADFP **does not require any additional or specialized training stage for the victim (teacher) model**. As shown in Algorithm 1, ADFP is a purely *inference-time* sampling algorithm. It computes a logit perturbation on the fly using a proxy model during the decoding step. The teacher model's weights (and proxy-model’s) remain completely frozen; no additional training costs are incurred.
>
> Furthermore, the verification stage **does not rely on secrecy, uniqueness, or specific "trigger queries."** (only the watermark hash key is private). ADFP implants a statistical watermark across the natural distribution of the model's outputs and the detection is performed by evaluating the student's generation on a standard validation set (e.g., GSM8K questions) and computing a statistical p-value. There are no specific trigger words involved in the evaluation of the watermark.
>
> **W4 (Baselines: TRAP, LLMmap, EditMF)**
>
> We appreciate the literature recommendations. However, TRAP, LLMmap, and EditMF address fundamentally different threat models than ADFP. TRAP and LLMmap are *active probing* techniques to identify frozen black-box APIs, while EditMF injects fictitious knowledge directly into weights via *model editing*.
>
> Conversely, ADFP targets **generative knowledge distillation**, where an attacker trains a *new* student model using teacher-generated text. The fingerprint must survive the student's fine-tuning. Red-Green-List (RGL) watermarking is currently the standard baseline empirically proven to possess this radioactivity property necessary for fingerprinting \[1\].
>
> *\[1\] Sander et al., 2024\. "Watermarking makes language models radioactive."*
>
> **W3 (MoE Architectures and Tokenization Schemes)**
>
> To evaluate ADFP on Mixture-of-Experts architectures and radically different tokenization schemes, we conduct new full parameter fine-tuning experiments on GSM8K using a sparse MoE student model (`allenai/OLMoE-1B-7B-0125`). Crucially, this OLMoE student (as well as `Llama-3.2-3B` that we already consider in the paper) uses a completely different tokenizer from our teacher model, directly testing cross-tokenizer robustness.
>
> We present the mean natural log p-value $\\pm 1.96 \\times \\text{standard error of mean}$ computed across 10 independent runs for the most challenging **closed & unsupervised** setting:
>
> | Student FT Setting | Student Model | ADFP | RGL |
> | :---- | :---- | :---- | :---- |
> | Full FT | OLMoE-1B-7B | $-13.704 \\pm 3.288$ | $-9.110 \\pm 3.439$ |
>
> The results show that the ADFP fingerprint robustly transfers to the MoE student and remains statistically stronger than the RGL baseline, even under cross-tokenizer discrepancies.
>
> **Q1 (Sample Complexity)**
>
> The theoretical results of this paper focus on the validity of the p-value computed by Algorithm 2\. We do not intend to provide a theoretical bound on the number of queries. Empirically, our GSM8K experiments demonstrate that robust detection can be achieved with a distillation dataset of approximately 7,400 queries across various distillation recipes and student models.
>
> **Q2 (Quantized Training)**
>
> Regarding quantization, we first clarify the threat model: the attacker only has black-box API access to the victim and receives discrete text tokens, which cannot be quantized. Moreover, the attacker lacks knowledge of the victim's quantization level or serving details (analogous to querying proprietary APIs like ChatGPT). We therefore interpret this question as whether the fingerprint survives if the attacker uses a heavily quantized training recipe (e.g., 4-bit quantization) for the *student model* during distillation.
>
> To address this, we evaluate ADFP under scenarios where the attacker fine-tunes the student using 4-bit and 8-bit QLoRA. Reusing the identical teacher traces from our original Figure 5, the results for the most challenging **closed & unsupervised** setting are as follows:
>
> | Student FT Setting | Student Model | ADFP | RGL |
> | :---- | :---- | :---- | :---- |
> | QLoRA 4-bit | Qwen2.5-3B | $-4.000 \\pm 1.209$ | $-0.556 \\pm 0.518$ |
> | QLoRA 8-bit | Qwen2.5-3B | $-3.533 \\pm 1.178$ | $-0.661 \\pm 0.643$ |
>
> As shown, the fingerprint signal of ADFP remains strong and statistically significant even when the attacker distills the outputs into a heavily quantized student model.
>
> We hope these clarifications and new experiments resolve your concerns, and we kindly ask you to reevaluate our submission in light of this new evidence.

---

> > ### Author Rebuttal · Reviewer_9UEQ · 2026-04-03
> >
> > I thank the authors for the rebuttal and my concerns are partially resolved. However, I am not totally convinced by authors' explaination for W4 (Baselines: TRAP, LLMmap, EditMF).
> >
> > In my understanding, all these methods are fingerprinting for protecting intellectural propoerty of LLMs while the authors argue that they are designed for different threat models. Does author imply other baselines, such as TRAP and EditMF, are not robust in the distillation scenarios? If so, I believe some simple experimental evidence should be provided. Otherwise, I think the comparison with baselines is not comprehensive.

---

> > > ### Author Response · Authors · 2026-04-03
> > >
> > > We sincerely thank you for engaging with our rebuttal. We are glad that our previous response appears to have addressed your concerns regarding the training cost (W1), the trigger queries (W2), the generalization to MoE architectures and different tokenizers (W3), the sample complexity (Q1), and the robustness to quantized distillation (Q2). We appreciate the opportunity to clarify the remaining point regarding the baselines (W4).
> > >
> > > We agree that methods such as TRAP, LLMmap, and EditMF are relevant prior works under the broad umbrella of LLM fingerprinting. However, our paper studies the more specific problem of **distillation fingerprinting**. In our setting, the teacher fingerprints its ordinary outputs so that the resulting statistical structure in the generated text can later be detected in a downstream student model after fine-tuning / distillation. This is why our direct empirical baseline is red-and-green-list watermarking: it operates in the same output-side distillation detection framework and is therefore directly comparable.
> > >
> > > **Distillation Fingerprinting (ADFP & RGL)**
> > >
> > > * **Scenario:** A model owner deploys a teacher model and wishes to detect whether a third party later fine-tunes a student model on the teacher’s outputs.
> > > * **Motivation:** The goal is to detect **distillation from outputs**.
> > > * **Technology:** The fingerprint is embedded into the **generated text at inference time** and is designed to transfer through the student’s training process.
> > > * **Why it is directly comparable:** Both ADFP and RGL are output-side methods with the same basic detect-after-distillation goal.
> > >
> > > **Active Probing / Model Identification (TRAP & LLMmap)**
> > >
> > > * **Scenario:** A third-party system exposes an LLM-backed service, and an auditor wants to identify which backend model is being used.
> > > * **Motivation:** The goal is to verify the identity or version of a deployed model.
> > > * **Technology:** These methods rely on carefully designed probe queries and the resulting responses to identify the underlying model.
> > > * **Distinction:** They are not designed to study whether a fingerprint transfers through ordinary training data into a separately fine-tuned student model. Therefore, they are not apples-to-apples empirical baselines for the distillation-fingerprinting setting in our paper.
> > >
> > > **Weight-Edited Ownership Fingerprints (EditMF)**
> > >
> > > * **Scenario:** A model developer modifies model parameters to embed an ownership signal before release.
> > > * **Motivation:** The goal is to verify ownership of a suspect model.
> > > * **Technology:** The signal is planted through **weight editing**, and verification is performed by querying for that embedded behavior.
> > > * **Distinction:** This is methodologically different from detecting whether a student model learned a fingerprint from ordinary teacher outputs during distillation. Because EditMF relies on hidden backdoors, normal API usage would not trigger the embedded behavior. Consequently, the fingerprint would never appear in the training data, making it inherently impossible to transfer to the student via distillation. Thus, while relevant as a neighboring line of work, it is not a direct empirical baseline for our setting.
> > >
> > > Ultimately, the suggested methods address neighboring but different threat models, such as deployed-model identification or ownership verification, whereas our paper studies **statistical fingerprint transfer through distillation**. For this reason, we do not believe they are direct empirical baselines for the problem considered here.
> > >
> > > At the same time, we agree that this distinction can be stated more clearly, and we will revise the related work and discussion to make the threat-model separation explicit. We hope this clarifies why red-and-green-list watermarking is the direct comparison in our experiments, and if this was the main remaining concern, we would be grateful if you would reconsider the score. Thank you again for your engagement.

---

### Official Review · Reviewer_zckR · 2026-03-13

**Soundness:** 3
**Presentation:** 3
**Significance:** 2
**Originality:** 3
**Overall Recommendation:** 4
**Confidence:** 3

**Summary:**

This paper studies the problem of detecting model distillation through fingerprinting. The authors propose Antidistillation Fingerprinting (ADFP), a watermarking scheme designed to remain detectable even after a student model is trained on outputs generated by a teacher model. Unlike prior heuristic watermarking approaches such as red–green list perturbations, ADFP derives its perturbation from the gradient of the student distillation objective, aligning the fingerprint signal with the student’s learning dynamics. The resulting method perturbs the teacher logits in a way that maximizes the learnability of the fingerprint by downstream student models. The paper also provides a statistical detection procedure with p-value guarantees and evaluates the method on GSM8K and OASST1. Experimental results show that ADFP achieves stronger fingerprint detectability than existing watermarking approaches while maintaining comparable generation quality.

**Compliance With Llm Reviewing Policy:**

Affirmed.

**Key Questions For Authors:**

Please refer to Weakness.

**Limitations:**

Yes

**Strengths And Weaknesses:**

Strengths:

- Detecting whether a model has been distilled from proprietary models is an important issue for intellectual property protection in large language models.

- The paper provides a theoretical derivation connecting the perturbation design with the learning dynamics of student models.

- Experiments on GSM8K and OASST1 demonstrate that ADFP achieves a better trade-off between fingerprint detectability and generation quality than red–green watermarking.

Weakness:

- Experiments are conducted on a small number of tasks (primarily GSM8K and OASST1). It is unclear whether the proposed fingerprinting scheme generalizes across other generation domains or model architectures.

- The paper mainly evaluates standard distillation settings where student models are trained directly on teacher outputs. However, realistic distillation pipelines may involve data transformations such as paraphrasing, filtering, or mixing with other datasets. These transformations could potentially weaken the fingerprint signal, but their impact is not evaluated in the paper. As a result, the robustness of the proposed fingerprinting scheme under such adversarial data transformations remains unclear.

- The method is motivated as aligning the fingerprint with the student’s learning dynamics, but the experiments keep the student fine-tuning recipe largely fixed to LoRA-based training for one epoch. The paper does not analyze how the fingerprint transfer behaves under different optimizers, PEFT methods, or alternative student training recipes

---

> ### Author Rebuttal · Authors · 2026-03-30
>
> We sincerely thank you for your constructive review and valuable feedback. We are greatly encouraged that you appreciate our "theoretical derivation connecting the perturbation design with the learning dynamics" and recognize that ADFP achieves a "better trade-off between fingerprint detectability and generation quality."
>
> **W1 (Generalization Across Domains and Architectures)**
>
> To address your concern, we conduct extensive new experiments on the strictly structured **MBPP code generation benchmark** using the standard execution-based function-completion setup. We use `Qwen2.5-Coder-7B` as the teacher, `Qwen2.5-Coder-3B` as the proxy, and fine-tune both `Qwen2.5-Coder-3B` and a cross-architecture `Llama-3.2-3B` as students. We present the results for the most challenging and realistic **closed & unsupervised** setting below.
>
> [Figure: MBPP Closed Unsupervised (Qwen2.5-Coder-3B Student)](https://ibb.co/CKWw0Bwy)
> [Figure: MBPP Closed Unsupervised (Llama-3.2-3B Student)](https://ibb.co/B24cpmXr)
>
> Like in the mathematical reasoning and conversational fine-tuning we test in the paper, ADFP achieves a Pareto improvement over red-and-green-list for both settings. This concurs with our intuition. As illustrated in our conceptual diagram (Figure 1), ADFP does not blindly boost the probability of tokens with initially low logits. By computing perturbations based on the proxy model's Jacobian, syntactically invalid tokens naturally receive near-zero perturbations. Thus, ADFP selectively amplifies valid, high-likelihood tokens, granting it a significant advantage in preserving strict syntax.
>
> **W3 (Alternative Student Fine-Tuning Recipes)**
>
> To verify if the fingerprint survives alternative fine-tuning strategies, we conduct new experiments on GSM8K by reusing the exact same teacher reasoning traces from the experiments in our original Figure 5, where we fix $\\lambda=256$ for ADFP and $\\delta=7$ for RGL (yielding a similar trace quality). We apply heavily modified student fine-tuning recipes, including full parameter fine-tuning, QLoRA (4-bit and 8-bit), and using a Mixture-of-Experts student model (`allenai/OLMoE-1B-7B-0125`).
>
> We present the mean natural log p-value $\\pm 1.96 \\times \\text{standard error of mean}$ computed across 10 independent runs for the most challenging and realistic **closed & unsupervised** setting (more negative is stronger fingerprint effect):
>
> | Student FT Setting | Student Model | ADFP | RGL |
> | :---- | :---- | :---- | :---- |
> | Original LoRA | Qwen2.5-3B | $-3.478 \\pm 1.206$ | $-1.740 \\pm 1.477$ |
> | QLoRA 8-bit | Qwen2.5-3B | $-3.533 \\pm 1.178$ | $-0.661 \\pm 0.643$ |
> | QLoRA 4-bit  | Qwen2.5-3B | $-4.000 \\pm 1.209$ | $-0.556 \\pm 0.518$ |
> | Full FT | Qwen2.5-3B | $-1.871 \\pm 1.456$ | $-0.281 \\pm 0.220$ |
> | Full FT (3 epochs) | Qwen2.5-3B | $-8.239 \\pm 2.805$ | $-1.601 \\pm 0.655$ |
> | Full FT | OLMoE-1B-7B | $-13.704 \\pm 3.288$ | $-9.110 \\pm 3.439$ |
>
> The results show that, ADFP fingerprint survives different fine-tuning recipes of the student, and remains statistically stronger than the RGL baseline across all dense, quantized, and sparse MoE student regimes tested. We note that for the same number of gradient steps, full fine-tuning is the most resistant to the fingerprints. But as the number of gradient steps (i.e., epochs) increases, the fingerprint effect becomes more prominent.
>
> **W2 (Robustness against Adversarial Data Transformations)**
>
> Regarding data transformations, we evaluated data mixing **data mixing** (where the attacker mixes fingerprinted data with clean, non-fingerprinted data) in Section 5.3 (Figure 5\) of the original manuscript. The results demonstrate that ADFP is resilient to this dilution. For transformations like filtering and paraphrasing, we acknowledge that this is a limitation, primarily because there is no standardized or predictable pipeline for how an attacker might adversarially process the distillation data, making it difficult to design a fixed, meaningful empirical setup. However, our mixing ablation in Figure 5 serves as a indicator that the statistical biases implanted by ADFP exhibit resistance to post-processing and data noise.
>
> More fundamentally, robustness against aggressive paraphrasing remains a severe, open challenge even for standard text watermarking, where maintaining rigorous statistical p-values after semantic rewrites is already difficult. Antidistillation fingerprinting poses a strictly harder problem, as the signal must not only survive the text transformation but also successfully transfer through the subsequent gradient updates of the student model. We have included a discussion of these adversarial transformations as a critical boundary of current fingerprinting mechanisms and a direction for future work.
>
> We hope these new experiments and discussions resolve your concerns regarding generalization and robustness, and we would appreciate it if you could reevaluate the paper according to our response.

---

> > ### Author Rebuttal · Reviewer_zckR · 2026-04-03
> >
> > Thank you for the thorough and well-executed response.
> >
> > Overall, my concerns have been largely addressed. I will keep my score unchanged for now.

---

> > > ### Author Response · Authors · 2026-04-03
> > >
> > > Thank you for engaging with our rebuttal and for your kind words about our updated materials. We are very glad to hear that you found the additional experiments and discussion thorough and well executed, and that your concerns have been fully resolved.

---

### Official Review · Reviewer_mgRf · 2026-03-13

**Soundness:** 3
**Presentation:** 3
**Significance:** 4
**Originality:** 2
**Overall Recommendation:** 5
**Confidence:** 4

**Summary:**

This work proposes applying antidistillation sampling to a watermarking/fingerprinting scenario, where the teacher’s watermarking scheme is optimized to improve the detection rate of text generated by a student model.

**Compliance With Llm Reviewing Policy:**

Affirmed.

**Final Justification:**

They addressed my concerns. I keep my positive score.

**Key Questions For Authors:**

1.	When the proxy model does not match the actual student model, the performance of the proposed method degrades. Have you investigated which model types or families generally serve as better proxy models?

2.	Robustness is crucial for practical watermarking systems. Have you evaluated the robustness of the proposed method against common watermark removal attacks?

3.	How scalable is this optimization-based approach? In particular, is it possible to adapt existing advanced watermarking methods to the antidistillation setting?

**Limitations:**

Yes

**Strengths And Weaknesses:**

**Soundness**

The paper provides a clear and well-structured formalization of the antidistillation fingerprinting problem, and the optimization-based methodology is well motivated. The authors introduce a simple yet effective approximation to address the computational challenges of the algorithm, and the experimental results demonstrate promising antidistillation performance.

However, the isotropic approximation appears overly idealized. While the algorithm itself is intuitive, its connection to the theoretical analysis is somewhat weak. Another concern is the substantial trade-off observed between generation quality and fingerprinting effectiveness.



**Presentation**

The paper is clearly written and generally easy to follow.



**Significance**

Watermarking is a critical technique for establishing the provenance of LLM-generated content, and disputes related to model distillation have recently attracted increasing attention. In this context, antidistillation fingerprinting is highly valuable for both academia and industry. As one of the first works addressing this problem, the paper opens a promising new research direction.



**Originality**

The main technique used for antidistillation fingerprinting is derived from antidistillation sampling, making the contribution primarily incremental in terms of novelty.

---

> ### Author Rebuttal · Authors · 2026-03-30
>
> We sincerely thank you for your supportive review and valuable feedback. We are greatly encouraged that you find our formalization "clear and well-structured," the methodology "well motivated," and recognize that antidistillation fingerprinting is "highly valuable for both academia and industry."
>
> **Originality and Contributions**
>
> While we naturally draw inspiration from the gradient-based framework of antidistillation sampling (ADS), our work makes independent and non-trivial contributions to a fundamentally different problem of fingerprinting. As you insightfully noted, this is "one of the first works addressing this problem," which we believe is a significant conceptual contribution in itself.
>
> Technically, we are the first to mathematically formalize the model fingerprinting problem. Furthermore, while the theoretical connection involves idealized assumptions (as in many deep learning optimization frameworks), our proposed Jacobian approximation is a critical technical novelty. It makes the fingerprinting objective computationally tractable, and as our extensive empirical evaluations demonstrate, it is highly effective in practice.
>
> **Q1 (Proxy Model Selection)**
>
> While proxy-student architecture mismatch causes some expected degradation, the most critical factor for selecting an effective proxy model is actually its alignment with the **teacher** model. It is highly beneficial for the proxy and the teacher to share the same (or at least a very similar) tokenizer. Because ADFP operates by perturbing the teacher's output logits, a shared tokenization space ensures that the proxy's gradient calculations map to the teacher's vocabulary without misalignment or interpolation errors. In our experiments, using a smaller model from the same family as the teacher (e.g., `Qwen2.5-3B` proxy for a `DeepSeek-R1-Distill-Qwen-7B` teacher) serves as a practical choice.
>
> **Q2 (Robustness against Watermark Removal Attacks)**
>
> Regarding common watermark removal attacks, we would first like to gently note that we have evaluated **data mixing** (where the attacker mixes fingerprinted data with clean, non-fingerprinted data) in Section 5.3 (Figure 5\) of the original manuscript. The results demonstrate that ADFP is resilient to this dilution.
>
> For transformations like filtering and paraphrasing, we acknowledge that this is a limitation, primarily because there is no standardized or predictable pipeline for how an attacker might adversarially process the distillation data, making it difficult to design a fixed, meaningful empirical setup. However, our mixing ablation in Figure 5 serves as a strong indicator that the statistical biases implanted by ADFP exhibit resistance to post-processing and data noise.
>
> More fundamentally, robustness against aggressive paraphrasing remains a severe, open challenge even for standard text watermarking, where maintaining rigorous statistical p-values after semantic rewrites is already difficult. Antidistillation fingerprinting poses a strictly harder problem, as the signal must not only survive the text transformation but also successfully transfer through the subsequent gradient updates of the student model. We have included a discussion of these adversarial transformations as a critical boundary of current fingerprinting mechanisms and a direction for future work.
>
> **Q3 (Scalability and Adapting Advanced Watermarks)**
>
> Regarding computational scalability, our proposed Jacobian approximation specifically addresses the computational bottleneck, reducing the additional computational cost to only the autoregressive generation of a proxy model, which is usually much smaller than the teacher.
>
> Regarding adapting to existing advanced watermarking methods, ADFP is fundamentally a general optimization framework. The per-step loss function $L$ can theoretically be any differentiable test statistic or expected objective defined over the current token and context. As long as the detection objective for the watermark can be formulated as a differentiable expected loss, we can apply ADFP’s idea to obtain a fingerprinting scheme.
>
> We hope our response has addressed your questions, and we thank you again for your strong support of our work.

---

> > ### Author Rebuttal · Reviewer_mgRf · 2026-04-04
> >
> > Thanks for your rebuttal. I'll maintain my score.

---

> > > ### Author Response · Authors · 2026-04-04
> > >
> > > Thank you for taking the time to review our rebuttal and for confirming that your concerns have been fully resolved. We appreciate your constructive feedback and your continued support.

---

### Decision · Program_Chairs · 2026-04-30

**Decision:**

Accept (regular)

**Comment:**

The submission received the comments of four reviewers, who respectively rated the scores 5, 4, 4, 4. The initial concerns focus on the idealized approximation and weak theoretical connection, the limited experiments about dataset and baseline dimensions, as well as the unclear difference clarification compared with other protection methods. During the rebuttal, most of concerns are well addressed as the acknowledgement of three reviewers, while one reviewer is still confused by the scope difference compared with previous protection methods.

AC has checked the rebuttal between the reviewers and the authors, and for the remaining concerns about the scope difference, AC considered that can support the difference clarification, but suggest the authors to include the difference discussion into the revision. Overall, the setting of this submission is timely about the IP protection in distillation of LLMs, and the method is novel. Given the decent score on average and the good rebuttal support, AC tends to recommending "Accept". Hope the authors carefully take the reviewers' constructive suggestion into the final revision to improve the readability and logic completeness.